# Speedup Matrix Completion with Side Information: Application to Multi-Label Learning

**Miao Xu**[1]        **Rong Jin**[2]        **Zhi-Hua Zhou**[1]

[1]National Key Laboratory for Novel Software Technology,
Nanjing University, Nanjing 210023, China
[2]Department of Computer Science and Engineering,
Michigan State University, East Lansing, MI 48824

`{xum, zhouzh}@lamda.nju.edu.cn`    `rongjin@cse.msu.edu`

## Abstract

In standard matrix completion theory, it is required to have at least $O(n \ln^2 n)$ observed entries to perfectly recover a low-rank matrix $M$ of size $n \times n$, leading to a large number of observations when $n$ is large. In many real tasks, side information in addition to the observed entries is often available. In this work, we develop a novel theory of matrix completion that explicitly explore the side information to reduce the requirement on the number of observed entries. We show that, under appropriate conditions, with the assistance of side information matrices, the number of observed entries needed for a perfect recovery of matrix $M$ can be dramatically reduced to $O(\ln n)$. We demonstrate the effectiveness of the proposed approach for matrix completion in transductive incomplete multi-label learning.

## 1    Introduction

Matrix completion concerns the problem of recovering a low-rank matrix from a limited number of observed entries. It has broad applications including collaborative filtering [35], dimensionality reduction [41], multi-class learning [4, 31], clustering [15, 42], etc. Recent studies show that, with a high probability, we can efficiently recover a matrix $M \in \mathbb{R}^{n \times m}$ of rank $r$ from $O(r(n+m)\ln^2(n+m))$ observed entries when the observed entries are uniformly sampled from $M$ [11, 12, 34].

Although the sample complexity for matrix completion, i.e., the number of observed entries required for perfectly recovering a low rank matrix, is already near optimal (up to a logarithmic factor), its linear dependence on $n$ and $m$ requires a large number of observations for recovering large matrices, significantly limiting its application to real-world problems. Moreover, current techniques for matrix completion require solving an optimization problem that can be computationally prohibitive when the size of the matrix is very large. In particular, although a number of algorithms have been developed for matrix completion [10, 22, 23, 25, 27, 28, 39], most of them require updating the full matrix $M$ at each iteration of optimization, leading to a high computational cost and a large storage requirement when both $n$ and $m$ are large. Several recent efforts [5, 19] try to address this issue, at a price of losing performance guarantee in recovering the target matrix.

On the other hand, in several applications of matrix completion, besides the observed entries, side information is often available that can potentially benefit the process of matrix completion. Below we list a few examples:

- *Collaborative filtering* aims to predict ratings of individual users based on the ratings from other users [35]. Besides the ratings provided by users, side information, such as the textual description of items and the demographical information of users, is often available and can be used to facilitate the prediction of missing ratings.

- *Link prediction* aiming to predict missing links between users in a social network based on the existing ones can be viewed as a matrix completion problem [20], where side information, such as attributes of users (e.g., browse patterns and interaction among users), can be used to assist completing the user-user-link matrix.

Although several studies exploit side information for matrix recovery [1, 2, 3, 16, 29, 32, 33], most of them focus on matrix factorization techniques, which usually result in non-convex optimization problems without guarantee of perfectly recovering the target matrix. In contrast, matrix completion deals with convex optimization problems and perfect recovery is guaranteed under appropriate conditions.

In this work, we focus on exploiting side information to improve the sample complexity and scalability of matrix completion. We assume that besides the observed entries in the matrix $M$, there exist two *side information matrices* $A \in \mathbb{R}^{n \times r_a}$ and $B \in \mathbb{R}^{m \times r_b}$, where $r \leq r_a \leq n$ and $r \leq r_b \leq m$. We further assume the target matrix and the side information matrices share the same latent information; that is, the column and row vectors in $M$ lie in the subspaces spanned by the column vectors in $A$ and $B$, respectively. Unlike the standard theory of matrix completion that needs to find the optimal matrix $M$ of size $n \times m$, our optimization problem is reduced to searching for an optimal matrix of size $r_a \times r_b$, making the recovery significantly more efficient both computationally and storage wise provided $r_a \ll n$ and/or $r_b \ll m$. We show that, with the assistance of side information matrices, with a high probability, we can perfectly recover $M$ with $O(r(r_a + r_b) \ln(r_a + r_b) \ln(n + m))$ observed entries, a sample complexity that is sublinear in $n$ and $m$.

We demonstrate the effectiveness of matrix completion with side information in transductive incomplete multi-label learning [17], which aims to assign multiple labels to individual instances in a transductive learning setting. We formulate transductive incomplete multi-label learning as a matrix completion problem, i.e., completing the instance-label matrix based on the observed entries that correspond to the given label assignments. Both the feature vectors of instances and the class correlation matrix can be used as side information. Our empirical study shows that the proposed approach is particularly effective when the number of given label assignments is small, verifying our theoretical result, i.e., side information can be used to reduce the sample complexity.

The rest of the paper is organized as follows: Section 2 briefly reviews some related work. Section 3 presents our main contribution. Section 4 presents our empirical study. Finally Section 5 concludes with future issues.

## 2    Related work

**Matrix Completion**    The objective of matrix completion is to fill out the missing entries of a matrix based on the observed ones. Early work on matrix completion, also referred to as maximum margin matrix factorization [37], was developed for collaborative filtering. Theoretical studies show that, it is sufficient to perfectly recover a matrix $M \in \mathbb{R}^{n \times m}$ of rank $r$ when the number of observed entries is $O(r(n + m) \ln^2(n + m))$ [11, 12, 34]. A more general matrix recovery problem, referred to as matrix regression, was examined in [30, 36]. Unlike these studies, our proposed approach reduces the sample complexity with the help of side information matrices.

Several computational algorithms [10, 22, 23, 25, 27, 28, 39] have been developed to efficiently solve the optimization problem of matrix completion. The main problem with these algorithms lies in the fact that they have to explicitly update the full matrix of size $n \times m$, which is expensive both computationally and storage wise for large matrices. This issue has been addressed in several recent studies [5, 19], where the key idea is to store and update the low rank factorization of the target matrix. A preliminary convergence analysis is given in [19], however, none of these approaches guarantees *perfect* recovery of the target matrix, even with significantly large number of observed entries. In contrast, our proposed approach reduces the computational cost by explicitly exploring the side information matrices and still delivers the promise of perfect recovery.

Several recent studies involve matrix recovery with side information. [2, 3, 29, 33] are based on graphical models by assuming special distribution of latent factors; these algorithms, as well as [16] and [32], consider side information in matrix factorization. The main limitation lies in the fact that they have to solve non-convex optimization problems, and do not have theoretical guarantees on matrix recovery. Matrix completion with infinite dimensional side information was exploited in [1],

yet lacking guarantee of perfect recovery. In contrast, our work is based on matrix completion theory that deals with a general convex optimization problem and is guaranteed to make a perfect recovery of the target matrix.

**Multi-label Learning**   Multi-label learning allows each instance to be assigned to multiple classes simultaneously, making it more challenging than multi-class learning. The simplest approach for multi-label learning is to train one binary model for each label, which is also referred to as **BR** (Binary Relevance) [7]. Many advanced algorithms have been developed to explicitly explore the dependence among labels ( [44] and references therein).

In this work, we will evaluate our proposed approach by *transductive incomplete multi-label learning* [17]. Let $X = (\mathbf{x}_1, \ldots, \mathbf{x}_n)^\top \in \mathbb{R}^{n \times d}$ be the feature matrix with $\mathbf{x}_i \in \mathbb{R}^d$, where $n$ is the number of instances and $d$ is the dimension. Let $\mathcal{C}_1, \ldots, \mathcal{C}_m$ denote the $m$ labels, and let $T \in \{-1, +1\}^{n \times m}$ be the instance-label matrix, where $T_{i,j} = +1$ when $\mathbf{x}_i$ is associated with the label $\mathcal{C}_j$, and $T_{i,j} = -1$ when $\mathbf{x}_i$ is not associated with the label $\mathcal{C}_j$. Let $\Omega$ denote the subset of the observed entries in $T$ that corresponds to the given label assignments of instances. The objective of transductive incomplete multi-label learning is to predict the missing entries in $T$ based on the feature matrix $X$ and the given label assignments in $\Omega$. The main challenge lies in the fact that only a *partial* label assignment is given for each training instance. This is in contrast to many studies on common semi-supervised or transductive multi-label learning [18, 24, 26, 43] where each labeled instance receives a *complete* set of label assignments. This is also different from multi-label learning with weak labels [8, 38] which assumes that only the positive labels can be observed. Here we assume the observed labels can be either positive or negative.

In [17], a matrix completion based approach was proposed for transductive incomplete multi-label learning. To effectively exploit the information in the feature matrix $X$, the authors proposed to complete the matrix $T' = [X, T]$ that combines the input features with label assignments into a single matrix. Two algorithms **MC-b** and **MC-1** were presented there, differing only in the treatment of bias term, whereas the convergence of MC-1 was examined in [9]. The main limitation of both algorithms lies in their high computational cost when both the number of instances and features are large. Unlike MC-1 and MC-b, our proposed approach does not need to deal with the big matrix $T'$, and is computationally more efficient. Besides the computational advantage, we show that our proposed approach significantly improves the sample complexity of matrix completion by exploiting side information matrices.

## 3   Speedup Matrix Completion with Side Information

We first describe the framework of matrix completion with side information, and then present its theoretical guarantee and application to multi-label learning

### 3.1   Matrix Completion using Side Information

Let $M \in \mathbb{R}^{n \times m}$ be the target matrix of rank $r$ to be recovered. Without loss of generality, we assume $n \geq m$. Let $\lambda_k, k \in \{1, \ldots, r\}$ be the $k$th largest singular value of $M$, and let $\mathbf{u}_k \in \mathbb{R}^n$ and $\mathbf{v}_k \in \mathbb{R}^m$ be the corresponding left and right singular vectors, i.e., $M = U\Sigma V^\top$, where $\Sigma = diag(\lambda_1, \ldots, \lambda_r)$, $U = (\mathbf{u}_1, \ldots, \mathbf{u}_r)$ and $V = (\mathbf{v}_1, \ldots, \mathbf{v}_r)$.

Let $\Omega \subseteq \{1, \ldots, n\} \times \{1, \ldots, m\}$ be the subset of indices of observed entries sampled uniformly from all entries in $M$. Given $\Omega$, we define a linear operator $\mathcal{R}_\Omega(M) : \mathbb{R}^{n \times m} \mapsto \mathbb{R}^{n \times m}$ as

$$[\mathcal{R}_\Omega(M)]_{i,j} = \begin{cases} M_{i,j} & (i,j) \in \Omega \\ 0 & (i,j) \notin \Omega \end{cases}$$

Using $\mathcal{R}_\Omega(\cdot)$, the standard matrix completion problem is:

$$\min_{\tilde{M} \in \mathbb{R}^{n \times m}} \|\tilde{M}\|_{tr} \quad \text{s. t.} \quad \mathcal{R}_\Omega(\tilde{M}) = \mathcal{R}_\Omega(M), \tag{1}$$

where $\| \cdot \|_{tr}$ is the trace norm.

Let $A = (\mathbf{a}_1, \ldots, \mathbf{a}_{r_a}) \in \mathbb{R}^{n \times r_a}$ and $B = (\mathbf{b}_1, \ldots, \mathbf{b}_{r_b}) \in \mathbb{R}^{m \times r_b}$ be the side information matrices, where $r \leq r_a \leq n$ and $r \leq r_b \leq m$. Without loss of generality, we assume that $r_a \geq r_b$ and that

both $A$ and $B$ are orthonormal matrices, i.e., $\mathbf{a}_i^\top \mathbf{a}_j = \delta_{i,j}$ and $\mathbf{b}_i^\top \mathbf{b}_j = \delta_{i,j}$ for any $i$ and $j$, where $\delta_{i,j}$ is the Kronecker delta function that outputs 1 if $i = j$ and 0, otherwise. In case when the side information is not available, $A$ and $B$ will be set to identity matrix.

The objective is to complete a matrix $M$ of rank $r$ with the side information matrices $A$ and $B$. We make the following assumption in order to fully exploit the side information:

**Assumption A**: the column vectors in $M$ lie in the subspace spanned by the column vectors in $A$, and the row vectors in $M$ lie in the subspace spanned by the column vectors in $B$.

To understand the implication of this assumption, let us consider the problem of transductive incomplete multi-label learning [17], where the objective is to complete the instance-label matrix based on the observed entries corresponding to the given label assignments, and the side information matrices $A$ and $B$ are given by the feature vectors of instances and the label correlation matrix, respectively. Assumption **A** essentially implies that all the label assignments can be accurately predicted by a linear combination of feature vectors of instances.

Using Assumption **A**, we can write $M$ as $M = AZ_0B^\top$ and therefore, our goal is to learn $Z_0 \in \mathbb{R}^{r_a \times r_b}$. Following the standard theory for matrix completion [11, 12, 34], we can cast the matrix completion task into the following optimization problem:

$$\min_{Z \in \mathbb{R}^{r_a \times r_b}} \|Z\|_{tr} \quad \text{s. t.} \quad \mathcal{R}_\Omega(AZB^\top) = \mathcal{R}_\Omega(M). \tag{2}$$

Unlike the standard algorithm for matrix completion that requires solving an optimization problem involved matrix of $n \times m$, the optimization problem given in (2) only deals with a matrix $Z$ of $r_a \times r_b$, and therefore can be solved significantly more efficiently if $r_a \ll n$ and $r_b \ll m$.

## 3.2 Theoretical Result

We define $\mu_0$ and $\mu_1$, the coherence measurements for matrix $M$ as

$$
\begin{aligned}
\mu_0 &= \max\left(\frac{n}{r} \max_{1 \le i \le n} \|P_U \mathbf{e}_i\|^2, \frac{m}{r} \max_{1 \le j \le m} \|P_V \mathbf{e}_j\|^2 \right), \\
\mu_1 &= \max_{i,j} \frac{mn}{r}([UV^\top]_{i,j})^2,
\end{aligned}
$$

where $\mathbf{e}_i$ is the vector with the $i$th entry equal to 1 and all others equal to 0, and $P_U$ and $P_V$ project a vector onto the subspace spanned by the column vectors of $U$ and $V$, respectively. We also define the coherence measure for matrices $A$ and $B$ as

$$\mu_{AB} = \max\left(\max_{1 \le i \le n} \frac{n\|A_{i,*}\|^2}{r_a}, \max_{1 \le j \le m} \frac{m\|B_{j,*}\|^2}{r_b} \right),$$

where $A_{i,*}$ and $B_{i,*}$ stand for the $i$th row of $A$ and $B$, respectively.

**Theorem 1.** *Let* $\mu = \max(\mu_0, \mu_{AB})$. *Define* $q_0 = \frac{1}{2}(1 + \log_2 r_a - \log_2 r)$, $\Omega_0 = \frac{128\beta}{3}\mu\max(\mu_1, \mu)r(r_a + r_b)\ln n$ *and* $\Omega_1 = \frac{8\beta}{3}\mu^2(r_a r_b + r^2)\ln n$. *Assume* $\Omega_1 \ge q_0 \Omega_0$. *With a probability at least* $1 - 4(q_0 + 1)n^{-\beta+1} - 2q_0 n^{-\beta+2}$, $Z_0$ *is the unique optimizer to the problem in (2) provided*

$$|\Omega| \ge \frac{64\beta}{3}\mu\max(\mu_1, \mu)(1 + \log_2 r_a - \log_2 r)r(r_a + r_b)\ln n.$$

Compared to the standard matrix completion theory [34], the side information matrices reduce sample complexity from $O(r(n+m)\ln^2(n+m))$ to $O(r(r_a + r_b)\ln(r_a + r_b)\ln n)$. When $r_a \ll n$ and $r_b \ll m$, the side information allows us significantly reduce the number of observed entries required for perfectly recovering matrix $M$. We defer the technical proof of Theorem 1 to the supplementary material due to page limit. Note that although we follow the framework of [34] for analysis, namely first proving the result under deterministic conditions, and then showing that the deterministic conditions hold with a high probability, our technical proof is quite different due to the involvement of side information matrices $A$ and $B$.

### 3.3 Application to Multi-Label Learning

Similar to the Singular Vector Thresholding (SVT) method [10], we approximate the problem in ( 2) by an unconstrained optimization problem, i.e.,

$$\min_{Z \in \mathbb{R}^{r_a \times r_b}} \mathcal{L}(Z) = \lambda \|Z\|_{tr} + \frac{1}{2} \left\| \mathcal{R}_\Omega(AZB^\top - M) \right\|_F^2, \tag{3}$$

where $\lambda > 0$ is introduced to weight the trace norm regularization term against the regression error. We develop an algorithm that exploits the smoothness of the loss function and therefore achieves $O(1/T^2)$ convergence, where $T$ is the number of iterations. Details of the algorithm can be found in the supplementary material. We refer to the proposed algorithm as **Maxide**.

For transductive incomplete multi-label learning, we abuse our notation by defining $n$ as the number of instances, $m$ as the number of labels, and $d$ as the dimensionality of input patterns. Our goal is to complete the instance-label matrix $M \in \mathbb{R}^{n \times m}$ by using (i) the feature matrix $X \in \mathbb{R}^{n \times d}$ and (ii) the observed entries $\Omega$ in $M$ (i.e., the given label assignments). We thus set the side information matrix $A$ to include the top left singular vectors of $X$, and $B = I$ to indicate that no side information is available for the dependence among labels. We note that the low rank assumption of instance-label matrix $M$ implies a linear dependence among the label prediction functions. This assumption has been explored extensively in the previous studies of multi-label learning [17, 21, 38].

## 4 Experiments

We evaluate the proposed algorithm for matrix completion with side information on both synthetic and real data sets. Our implementation is in Matlab except that the operation $\mathcal{R}_\Omega(L \times R)$ is implemented in C. All the results were obtained on a Linux server with CPU 2.53GHz and 48GB memory.

### 4.1 Experiments on Synthetic Data

To create the side information matrices $A$ and $B$, we first generate a random matrix $F \in \mathbb{R}^{n \times m}$, with each entry $F_{i,j}$ drawn independently from $\mathcal{N}(0, 1)$. Side information matrix $A$ includes the first $r_a$ left singular vectors of $F$, and $B$ includes the first $r_b$ right singular vectors. To create $Z_0$, we generate two Gaussian random matrices $Z_A \in \mathbb{R}^{r_a \times r}$ and $Z_B \in \mathbb{R}^{r_b \times r}$, where each entry is sampled independently from $\mathcal{N}(0, 1)$. The singular value decomposition of $AZ_A$ and $BZ_B$ is given by $AZ_A = U\Sigma_1 V_1^T$ and $BZ_B = V\Sigma_2 V_2^T$, respectively. We create a diagonal matrix $\Sigma \in \mathbb{R}^{r \times r}$, whose diagonal entries are drawn independently from $\mathcal{N}(0, 10^4)$. $Z_0$ is then given by $Z_0 = (Z_A \Sigma_1^\dagger (V_1^T)^\dagger) \Sigma (Z_B \Sigma_2^\dagger (V_2^T)^\dagger)^T$ where $\dagger$ denotes the pseudo inverse of a matrix. Finally, the target matrix $M$ is given by $M = AZ_0 B^\top$.

**Settings and Baselines**   Our goal is to show that the proposed algorithm is able to accurately recover the target matrix with significantly smaller number of entries and less computational time. In this study, we only consider square matrices (i.e., $m = n$), with $n = 1,000; 5,000; 10,000; 20,000; 30,000$ and rank $r = 10; 50; 100$. Both $r_a$ and $r_b$ of side information matrices are set to be $2r$, and $|\Omega|$, the number of observed entries, is set to be $r(2n - r)$, which is significantly smaller than the number of observed entries used in previous studies [10, 25, 27]. We repeat each experiment 10 times, and report the result averaged over 10 runs. We compare the proposed Maxide algorithm with three state-of-the-art matrix completion algorithms: Singular Vector Thresholding (**SVT**) [10], Fixed Point Bregman Iterative Method (**FPCA**) [27] and Augmented Lagrangian Method (**ALM**) [25]. In addition to these matrix completion methods, we also compare with a trace norm minimizing method (**TraceMin**) [6]. For all the baseline, we use the codes provided by their original authors with their default parameter settings.

**Results**   We measure the performance of matrix completion by the relative error $\|AZB^\top - M\|_F / \|M\|_F$ and report the results of both relative error and running time in Table 1. For TraceMin, we observe that for $n = 1,000$ and $r = 10$, it gives the result of $1.75 \times 10^{-7}$ within $2.94 \times 10^4$ seconds, which is really slow compared to our proposal. For $n = 1,000$ and $r = 50$, it gives no result within one week. In Table 1, we first observed that for all the cases, the relative error achieved

Table 1: Results on synthesized data sets. $n$ is the size of a squared matrix and $r$ is its rank. *Rate* is the number of observed entries divided by the size of the matrix, that is, $|\Omega|/(nm)$. *Time* measures the running time in seconds and *Relative error* measures $\|AFB^\top - M\|_F/\|M\|_F$. The best performance for each setting are bolded. We do not report the results for FPCA and SVT when $n \geq 5,000$ because they were unable to finish the computation after 50 hours.

| $n$ | $r$ | Rate | Alg. | Time | Relative error | Algo. | Time | Relative error |
|---|---|---|---|---|---|---|---|---|
| 1,000 | 10 | $1.99 \times 10^{-2}$ | Maxide | $\mathbf{1.89 \times 10^1}$ | $\mathbf{6.42 \times 10^{-7}}$ | FPCA | $5.55 \times 10^3$ | $8.79 \times 10^{-1}$ |
| | | | SVT | $3.23 \times 10^3$ | $8.76 \times 10^4$ | ALM | $2.92 \times 10^1$ | $8.46 \times 10^{-1}$ |
| | 50 | $9.75 \times 10^{-2}$ | Maxide | $\mathbf{6.44 \times 10^1}$ | $\mathbf{5.28 \times 10^{-8}}$ | FPCA | $7.60 \times 10^3$ | $5.53 \times 10^{-1}$ |
| | | | SVT | $3.51 \times 10^3$ | $2.77 \times 10^5$ | ALM | $7.72 \times 10^1$ | $5.58 \times 10^{-1}$ |
| | 100 | $1.900 \times 10^{-1}$ | Maxide | $1.94 \times 10^2$ | $\mathbf{1.91 \times 10^{-8}}$ | FPCA | $1.71 \times 10^4$ | $4.63 \times 10^{-1}$ |
| | | | SVT | $3.82 \times 10^3$ | $7.45 \times 10^4$ | ALM | $\mathbf{8.57 \times 10^1}$ | $3.59 \times 10^{-1}$ |
| 5,000 | 10 | $3.96 \times 10^{-3}$ | Maxide | $\mathbf{3.50 \times 10^1}$ | $\mathbf{6.38 \times 10^{-4}}$ | ALM | $1.24 \times 10^3$ | $9.07 \times 10^{-1}$ |
| | 50 | $1.99 \times 10^{-2}$ | Maxide | $\mathbf{4.56 \times 10^2}$ | $\mathbf{1.43 \times 10^{-7}}$ | ALM | $1.79 \times 10^3$ | $7.26 \times 10^{-1}$ |
| | 100 | $3.96 \times 10^{-2}$ | Maxide | $\mathbf{1.29 \times 10^3}$ | $\mathbf{2.44 \times 10^{-8}}$ | ALM | $2.14 \times 10^3$ | $5.51 \times 10^{-1}$ |
| 10,000 | 10 | $2.00 \times 10^{-3}$ | Maxide | $\mathbf{6.18 \times 10^1}$ | $\mathbf{1.63 \times 10^{-3}}$ | ALM | $7.16 \times 10^3$ | $9.10 \times 10^{-1}$ |
| | 50 | $9.98 \times 10^{-3}$ | Maxide | $\mathbf{8.39 \times 10^2}$ | $\mathbf{9.97 \times 10^{-2}}$ | ALM | $7.87 \times 10^3$ | $7.19 \times 10^{-1}$ |
| | 100 | $1.99 \times 10^{-2}$ | Maxide | $\mathbf{4.47 \times 10^3}$ | $\mathbf{1.67 \times 10^{-7}}$ | ALM | $9.50 \times 10^3$ | $6.41 \times 10^{-1}$ |
| 20,000 | 10 | $1.00 \times 10^{-3}$ | Maxide | $\mathbf{1.22 \times 10^2}$ | $\mathbf{3.54 \times 10^{-3}}$ | ALM | $3.62 \times 10^4$ | $9.49 \times 10^{-1}$ |
| | 50 | $4.99 \times 10^{-3}$ | Maxide | $\mathbf{2.16 \times 10^3}$ | $\mathbf{4.51 \times 10^{-4}}$ | ALM | $4.09 \times 10^4$ | $8.51 \times 10^{-1}$ |
| 30,000 | 10 | $6.67 \times 10^{-4}$ | Maxide | $\mathbf{4.37 \times 10^2}$ | $\mathbf{3.25 \times 10^{-3}}$ | ALM | $8.69 \times 10^4$ | $9.53 \times 10^{-1}$ |

by the baseline methods is $\Omega(1)$, implying that none of them is able to make accurate recovery of the target matrix given the small number of observed entries. In contrast, our proposed algorithm is able to recover the target matrix with small relative error. In addition, our proposed algorithm is computationally more efficient than the baseline methods. The improvement in computational efficiency becomes more significant for large matrices.

## 4.2 Application to Transductive Incomplete Multi-Label Learning

We evaluate the proposed algorithm for transductive incomplete multi-label learning on thirteen benchmark data sets, including eleven data sets for web page classification from "yahoo.com" [40], and two image classification data sets NUS-WIDE [14] and Flickr [45]. For the eleven "yahoo.com" data sets, the number of instances is $n = 5,000$ and the number of dimensions varies from 438 to 1,047, with the number of labels varies from 21 to 40. Detailed information of these eleven data sets can be found in [40]. For NUS-WIDE data set, we have $n = 209,347$ images each represented by a bag-of-words model with $d = 500$ visual words, and 81 labels. For the Flickr data set, we only keep the first 1,000 most popular keywords for labels, leaving us with $n = 565,444$ images, each represented by a $d = 297$-dimension vector.

**Settings and Baselines** For each data set, we randomly sample 10% instances for testing (unlabeled data) and use the remaining 90% data for training. No label assignment is provided for any test instance. To create partial label assignments for training data, for each label $C_j$, we expose the label assignment of $C_j$ for $\omega\%$ randomly sampled positive and negative training instances and keep the label assignment of $C_j$ unknown for the rest of the training instances. To examine the performance of the proposed algorithm, we vary the $\omega\%$ in the range $\{10\%, 20\%, 40\%\}$. We repeat each experiment 10 times, and report the result averaged over 10 trials. The regularization parameter $\lambda$ is selected from $2^{\{-10,-9,\dots,9,10\}}$ by cross validation on training data for smaller data sets and set as 1 for larger ones. Parameters $\gamma$ and $\epsilon$ are set to be 2 and $10^{-5}$, respectively, for the proposed algorithm, and the maximum number of iterations is set to be 100. The Average Precision [44], which measures the average number of relevant labels ranked before a particular relevant label, is computed over the test data (the metric on all the data is provided in the supplementary material) and used as our evaluation metric.

We compare the proposed Maxide method with MC-1 and MC-b, the state-of-the-art methods for transductive incomplete multi-label learning developed in [17]. In addition, we also compare with two reference methods for multi-label learning that train one binary classifier for each label; that is, the Binary Relevance method [7] based on Linear kernel (**BR-L**) and the method based on RBF kernel (**BR-R**), where the kernel width is set to 1. For the eleven data sets from "yahoo.com",

LIBSVM [13] is used by BR-L and BR-R to learn both a linear and nonlinear SVM classifier. For the two image data sets, due to their large size, only BR-L method is included in comparison and LIBLINEAR is used for the implementation of BR-L due to its high efficiency for large data sets. A similar strategy is used to determine the optimal $\lambda$ as our proposal.

**Results**    Table 2 summarizes the results on transductive incomplete multi-label learning. We observe that the proposed Maxide algorithm outperforms the baseline methods, for most setting on several data sets (e.g., Business, Education, and Recreation), and the improvements are significant. More impressively, for most data sets, the proposed algorithm is three order faster than MC-1 and MC-b. For the NUS-WIDE data set, none of MC-1 and MC-b, the two existing matrix completion based algorithms for transductive incomplete multi-label learning, is able to finish within one week. For the Flickr data set, MC-1 and MC-b are not runnable due to the out of memory problem. For the NUS-WIDE and Flickr data sets, our proposed Maxide method gets an average of more than $50\%$ improvement against BR-L, the only runnable baseline, on the Average Precision.

## 5    Conclusion

In this paper, we develop the theory of matrix completion with side information. We show theoretically that, with side information matrices $A \in \mathbb{R}^{n \times r_a}$ and $B \in \mathbb{R}^{m \times r_b}$, we can perfectly recover an $n \times m$ rank-$r$ matrix with only $O(r(r_a + r_b) \ln(r_a + r_b) \ln(n + m))$ observed entries, a significant improvement compared to the sample complexity $O(r(n + m) \ln^2(n + m))$ for the standard theory for matrix completion. We present the Maxide algorithm that can efficiently solve the optimization problem for matrix completion with side information. Empirical studies with synthesized data sets and transductive incomplete multi-label learning show the promising performance of the proposed algorithm.

**Acknowledgement**    This research was partially supported by 973 Program (2010CB327903), NSFC (61073097, 61273301), and ONR Award (N000141210431).

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

Table 2: Results on transductive incomplete multi-label learning. *Algo.* specifies the name of the algorithms. *Time* is the CPU time measured in seconds. *AP* is Average Precision measured based on test data; the higher the *AP*, the better the performance. $\omega\%$ represents the percentage of training instances with observed label assignment for each label. The best result and its comparable ones (pairwise single-tailed $t$-tests at 95% confidence level) are bolded.

| Data | Algo. | $\omega\% = 10\%$ | | $\omega\% = 20\%$ | | $\omega\% = 40\%$ | |
|---|---|---|---|---|---|---|---|
| | | time | AP | time | AP | time | AP |
| Arts | Maxide | $3.09 \times 10^0$ | **0.548** | $3.60 \times 10^0$ | **0.572** | $4.42 \times 10^0$ | **0.596** |
| | MC-b | $2.47 \times 10^4$ | 0.428 | $1.59 \times 10^4$ | 0.444 | $9.54 \times 10^3$ | 0.434 |
| | MC-1 | $2.39 \times 10^4$ | 0.430 | $2.05 \times 10^4$ | 0.494 | $1.27 \times 10^4$ | 0.473 |
| | BR-R | $1.63 \times 10^1$ | 0.540 | $2.98 \times 10^1$ | 0.563 | $5.71 \times 10^1$ | 0.574 |
| | BR-1 | $1.77 \times 10^1$ | 0.540 | $3.07 \times 10^1$ | 0.563 | $7.10 \times 10^1$ | 0.575 |
| Business | Maxide | $3.24 \times 10^0$ | **0.868** | $3.89 \times 10^0$ | **0.860** | $5.04 \times 10^0$ | **0.872** |
| | MC-b | $2.94 \times 10^4$ | 0.865 | $1.83 \times 10^4$ | 0.851 | $1.08 \times 10^4$ | 0.858 |
| | MC-1 | $3.25 \times 10^4$ | 0.865 | $2.18 \times 10^4$ | 0.855 | $1.21 \times 10^4$ | 0.862 |
| | BR-R | $1.02 \times 10^1$ | 0.846 | $1.78 \times 10^1$ | 0.841 | $3.32 \times 10^1$ | 0.854 |
| | BR-1 | $1.19 \times 10^1$ | 0.846 | $1.96 \times 10^1$ | 0.841 | $4.30 \times 10^1$ | 0.854 |
| Computers | Maxide | $4.67 \times 10^0$ | **0.635** | $5.81 \times 10^0$ | **0.660** | $7.79 \times 10^0$ | **0.675** |
| | MC-b | $5.58 \times 10^4$ | 0.597 | $3.38 \times 10^4$ | 0.599 | $1.87 \times 10^4$ | 0.604 |
| | MC-1 | $6.56 \times 10^4$ | 0.600 | $4.40 \times 10^4$ | 0.608 | $2.30 \times 10^4$ | 0.618 |
| | BR-R | $2.34 \times 10^1$ | 0.622 | $4.13 \times 10^1$ | 0.649 | $7.68 \times 10^1$ | 0.662 |
| | BR-1 | $2.70 \times 10^1$ | 0.621 | $4.50 \times 10^1$ | 0.648 | $8.25 \times 10^1$ | **0.661** |
| Education | Maxide | $4.40 \times 10^0$ | **0.566** | $5.41 \times 10^0$ | **0.604** | $6.73 \times 10^0$ | **0.618** |
| | MC-b | $3.82 \times 10^4$ | 0.472 | $2.40 \times 10^4$ | 0.478 | $1.32 \times 10^4$ | 0.474 |
| | MC-1 | $4.68 \times 10^4$ | 0.484 | $3.02 \times 10^4$ | 0.536 | $1.55 \times 10^4$ | 0.564 |
| | BR-R | $1.77 \times 10^1$ | 0.535 | $3.16 \times 10^1$ | 0.568 | $6.01 \times 10^1$ | 0.583 |
| | BR-1 | $1.94 \times 10^1$ | 0.535 | $3.28 \times 10^1$ | 0.568 | $6.94 \times 10^1$ | 0.583 |
| Entertainment | Maxide | $2.77 \times 10^0$ | **0.631** | $3.41 \times 10^0$ | **0.650** | $4.56 \times 10^0$ | **0.679** |
| | MC-b | $4.86 \times 10^4$ | 0.474 | $3.13 \times 10^4$ | 0.467 | $1.73 \times 10^4$ | 0.468 |
| | MC-1 | $4.40 \times 10^4$ | 0.489 | $4.15 \times 10^4$ | 0.492 | $2.27 \times 10^4$ | 0.578 |
| | BR-R | $1.89 \times 10^1$ | **0.628** | $3.38 \times 10^1$ | 0.638 | $6.47 \times 10^1$ | 0.668 |
| | BR-1 | $2.04 \times 10^1$ | **0.627** | $3.44 \times 10^1$ | 0.640 | $6.41 \times 10^1$ | 0.667 |
| Health | Maxide | $4.31 \times 10^0$ | **0.725** | $5.36 \times 10^0$ | **0.746** | $7.11 \times 10^0$ | **0.769** |
| | MC-b | $4.98 \times 10^4$ | 0.609 | $2.99 \times 10^4$ | 0.607 | $1.71 \times 10^4$ | 0.610 |
| | MC-1 | $5.82 \times 10^4$ | 0.626 | $3.82 \times 10^4$ | 0.632 | $2.03 \times 10^4$ | 0.645 |
| | BR-R | $2.03 \times 10^1$ | **0.725** | $3.61 \times 10^1$ | **0.742** | $6.83 \times 10^1$ | 0.757 |
| | BR-1 | $2.16 \times 10^1$ | **0.725** | $3.59 \times 10^1$ | 0.741 | $7.05 \times 10^1$ | 0.757 |
| Recreation | Maxide | $2.75 \times 10^0$ | **0.559** | $3.38 \times 10^0$ | **0.592** | $4.44 \times 10^0$ | **0.614** |
| | MC-b | $3.56 \times 10^4$ | 0.381 | $2.41 \times 10^4$ | 0.381 | $1.30 \times 10^4$ | 0.378 |
| | MC-1 | $3.48 \times 10^4$ | 0.381 | $3.25 \times 10^4$ | 0.430 | $1.90 \times 10^4$ | 0.421 |
| | BR-R | $1.97 \times 10^1$ | 0.548 | $3.48 \times 10^1$ | 0.574 | $6.53 \times 10^1$ | 0.596 |
| | BR-1 | $2.24 \times 10^1$ | 0.547 | $3.74 \times 10^1$ | 0.573 | $6.86 \times 10^1$ | 0.596 |
| Reference | Maxide | $5.11 \times 10^0$ | **0.635** | $6.47 \times 10^0$ | **0.666** | $8.49 \times 10^0$ | **0.696** |
| | MC-b | $9.38 \times 10^4$ | 0.565 | $5.38 \times 10^4$ | 0.561 | $2.75 \times 10^4$ | 0.575 |
| | MC-1 | $1.11 \times 10^5$ | 0.576 | $6.53 \times 10^4$ | 0.576 | $3.22 \times 10^4$ | 0.575 |
| | BR-R | $2.28 \times 10^1$ | **0.644** | $3.89 \times 10^1$ | **0.670** | $7.08 \times 10^1$ | **0.693** |
| | BR-1 | $2.71 \times 10^1$ | **0.644** | $4.34 \times 10^1$ | **0.669** | $7.48 \times 10^1$ | **0.692** |
| Science | Maxide | $6.21 \times 10^0$ | **0.513** | $7.67 \times 10^0$ | **0.543** | $1.02 \times 10^1$ | **0.568** |
| | MC-b | $6.80 \times 10^4$ | 0.395 | $3.94 \times 10^4$ | 0.403 | $2.06 \times 10^4$ | 0.394 |
| | MC-1 | $8.50 \times 10^4$ | 0.411 | $4.97 \times 10^4$ | 0.470 | $2.52 \times 10^4$ | 0.414 |
| | BR-R | $2.93 \times 10^1$ | **0.506** | $5.06 \times 10^1$ | **0.535** | $9.30 \times 10^1$ | 0.557 |
| | BR-1 | $3.60 \times 10^1$ | **0.506** | $5.91 \times 10^1$ | 0.535 | $1.04 \times 10^2$ | 0.557 |
| Social | Maxide | $7.18 \times 10^0$ | **0.721** | $9.09 \times 10^0$ | **0.748** | $1.21 \times 10^1$ | **0.754** |
| | MC-b | $1.71 \times 10^5$ | 0.582 | $9.65 \times 10^4$ | 0.595 | $4.56 \times 10^4$ | 0.594 |
| | MC-1 | $2.22 \times 10^5$ | 0.602 | $1.17 \times 10^5$ | 0.625 | $5.41 \times 10^4$ | 0.604 |
| | BR-R | $3.09 \times 10^1$ | **0.717** | $5.35 \times 10^1$ | 0.746 | $9.74 \times 10^1$ | 0.751 |
| | BR-1 | $3.71 \times 10^1$ | 0.717 | $6.00 \times 10^1$ | 0.746 | $1.02 \times 10^2$ | 0.751 |
| Society | Maxide | $3.69 \times 10^0$ | **0.580** | $4.54 \times 10^0$ | **0.594** | $5.80 \times 10^0$ | **0.616** |
| | MC-b | $4.75 \times 10^4$ | 0.550 | $2.93 \times 10^4$ | 0.545 | $1.62 \times 10^4$ | 0.552 |
| | MC-1 | $4.14 \times 10^4$ | 0.550 | $3.65 \times 10^4$ | 0.561 | $2.04 \times 10^4$ | 0.590 |
| | BR-R | $2.50 \times 10^1$ | **0.571** | $4.54 \times 10^1$ | **0.590** | $8.59 \times 10^1$ | **0.600** |
| | BR-1 | $2.84 \times 10^1$ | **0.572** | $4.92 \times 10^1$ | **0.590** | $9.58 \times 10^1$ | **0.601** |
| NUS-WIDE | Maxide | $1.47 \times 10^3$ | **0.513** | $2.10 \times 10^3$ | **0.519** | $3.53 \times 10^3$ | **0.522** |
| | BR-1 | $1.24 \times 10^2$ | 0.329 | $2.38 \times 10^2$ | 0.398 | $4.81 \times 10^2$ | 0.466 |
| Flickr | Maxide | $1.33 \times 10^4$ | **0.124** | $1.89 \times 10^4$ | **0.124** | $2.67 \times 10^4$ | **0.124** |
| | BR-1 | $2.48 \times 10^4$ | 0.064 | $4.74 \times 10^4$ | 0.074 | $1.11 \times 10^5$ | 0.077 |

[16] Y. Fang and L. Si. Matrix co-factorization for recommendation with rich side information and implicit feedback. In *Proceedings of the 2nd International Workshop on Information Heterogeneity and Fusion in*

*Recommender Systems*, 2011.

[17] A. Goldberg, X. Zhu, B. Recht, J.-M. Xu, and R. Nowak. Transduction with matrix completion: Three birds with one stone. In *NIPS*, 2010.

[18] Y. Guo and D. Schuurmans. Semi-supervised multi-label classification - a simultaneous large-margin, subspace learning approach. In *ECML*, 2012.

[19] P. Jain, P. Netrapalli, and S. Sanghavi. Provable matrix sensing using alternating minimization. In *NIPS Workshop on Optimization for Machine Learning*, 2012.

[20] A. Jalali, Y. Chen, S. Sanghavi, and H. Xu. Clustering partially observed graphs via convex optimization. In *ICML*, 2011.

[21] S. Ji, L. Tang, S. Yu, and J. Ye. Extracting shared subspace for multi-label classification. In *KDD*, 2008.

[22] S. Ji and J. Ye. An accelerated gradient method for trace norm minimization. In *ICML*, 2009.

[23] R. Keshavan, A. Montanari, and S. Oh. Matrix completion from a few entries. *IEEE TIT*, 56(6):2980–2998, 2010.

[24] X. Kong, M. Ng, and Z.-H. Zhou. Transductive multi-label learning via label set propagation. *IEEE TKDE*, 25(3):704–719, 2013.

[25] Z. Lin, M. Chen, L. Wu, and Y. Ma. The augmented lagrange multiplier method for exact recovery of corrupted low-rank matrices. Technical report, UIUC, 2009.

[26] Y. Liu, R. Jin, and L. Yang. Semi-supervised multi-label learning by constrained non-negative matrix factorization. In *AAAI*, 2006.

[27] S. Ma, D. Goldfarb, and L. Chen. Fixed point and bregman iterative methods for matrix rank minimization. *Mathematical Programming*, 128(1-2):321–353, 2011.

[28] R. Mazumder, T. Hastie, and R. Tibshirani. Spectral regularization algorithms for learning large incomplete matrices. *JMLR*, 11:2287–2322, 2010.

[29] A. Menon, K. Chitrapura, S. Garg, D. Agarwal, and N. Kota. Response prediction using collaborative filtering with hierarchies and side-information. In *KDD*, 2011.

[30] S. Negahban and M. Wainwright. Estimation of (near) low-rank matrices with noise and high dimensional scaling. *Annual of Statistics*, 39(2):1069–1097, 2011.

[31] G. Obozinski, B. Taskar, and M. Jordan. Joint covariate selection and joint subspace selection for multiple classification problems. *Statistics and Computing*, 20(2):231–252, 2010.

[32] W. Pan, E. Xiang, N. Liu, and Q. Yang. Transfer learning in collaborative filtering for sparsity reduction. In *AAAI*, 2010.

[33] I. Porteous, A. Asuncion, and M. Welling. Bayesian matrix factorization with side information and dirichlet process mixtures. In *AAAI*, 2010.

[34] B. Recht. A simpler approach to matrix completion. *JMLR*, 12:3413–3430, 2011.

[35] J. Rennie and N. Srebro. Fast maximum margin matrix factorization for collaborative prediction. In *ICML*, 2005.

[36] A. Rhode and A. Tsybakov. Estimation of high dimensional low rank matrices. *Annual of Statistics*, 39(2):887–930, 2011.

[37] N. Srebro, Jason D. Rennie, and T. Jaakkola. Maximum-margin matrix factorization. In *NIPS*. 2005.

[38] Y.-Y. Sun, Y. Zhang, and Z.-H. Zhou. Multi-label learning with weak label. In *AAAI*, 2010.

[39] K.-C. Toh and Y. Sangwoon. An accelerated proximal gradient algorithm for nuclear norm regularized linear least squares problems. *Pacific Journal of Optimization*, 2010.

[40] N. Ueda and K. Saito. Parametric mixture models for multi-labeled text. In *NIPS*, 2002.

[41] K. Weinberger and L. Saul. Unsupervised learning of image manifolds by semidefinite programming. *IJCV*, 70(1):77–90, 2006.

[42] J. Yi, T. Yang, R. Jin, A. Jain, and M. Mahdavi. Robust ensemble clustering by matrix completion. In *ICDM*, 2012.

[43] G. Yu, C. Domeniconi, H. Rangwala, G. Zhang, and Z. Yu. Transductive multi-label ensemble classification for protein function prediction. In *KDD*, 2012.

[44] M.-L. Zhang and Z.-H. Zhou. A review on multi-label learning algorithms. *IEEE TKDE*, in press.

[45] J. Zhuang and S. Hoi. A two-view learning approach for image tag ranking. In *WSDM*, 2011.

