[Supplementary Material]

# Supplementary Material of "Speedup Matrix Completion with Side Information: Application to Multi-Label Learning"

**Miao Xu**[1]        **Rong Jin**[2]        **Zhi-Hua Zhou**[1]

[1]National Key Laboratory for Novel Software Technology,
Nanjing University, Nanjing 210023, China
[2]Department of Computer Science and Engineering,
Michigan State University, East Lansing, MI 48824
{xum, zhouzh}@lamda.nju.edu.cn     rongjin@cse.msu.edu

In this supplementary material, we will prove Theorem 1 in Section 1 and state the details of **Maxide** algorithm in Section 2. We then give some additional experiments in Section 3.

## 1 Proof of Theorem 1

Our strategy is to first identify the deterministic conditions for $Z_0$ to be the unique minimizer for (2), which is given by Lemma 1. We then confirm that those deterministic conditions will hold with high probabilities in Lemma 7, Lemma 10 and Lemma 11. Finally, Theorem 1 can be proved using all these lemmas.

Before stating the detailed proof, we define a linear operator $P_T : \mathbb{R}^{n \times m} \mapsto \mathbb{R}^{n \times m}$ as follows: for any $F \in \mathbb{R}^{n \times m}$, $P_T$ maps $F$ to a new matrix $P_T(F)$ given by

$$P_T(F) = P_U F P_B + P_A F P_V - P_U F P_V, \tag{4}$$

where $P_U$, $P_V$, $P_A$ and $P_B$ project a vector onto the subspace spanned by the column vectors in $U$, $V$, $A$, and $B$, respectively. That is, if $U_A$ and $U_B$ are the left singular vectors of $A$ and $B$ respectively, then

$$
\begin{aligned}
P_U &= UU^T \in \mathbb{R}^{n \times n}, \\
P_V &= VV^T \in \mathbb{R}^{m \times m}, \\
P_A &= U_A U_A^T = AA^T \in \mathbb{R}^{n \times n}, \\
P_B &= U_B U_B^T = BB^T \in \mathbb{R}^{m \times m}.
\end{aligned}
$$

We note that the projection operator $P_T$ defined in (4) is different from that defined in [4] in that we restrict the left invariant space to $A$ and the right invariant space to $B$ due to our assumptions. Similarly, we define a linear operator $P_{T\perp}$ as

$$P_{T\perp}(F) = (P_A - P_U)F(P_B - P_V) = P_{A\perp} F P_{B\perp}.$$

For convenience, we rewrite the definitions of $\Omega_0$, $\Omega_1$ and $q_0$ in Theorem 1 here, which will make our future statement easier,

$$
\begin{aligned}
q_0 &= \frac{1}{2}(1 + \log_2 r_a - \log_2 r), \\
\Omega_0 &= \frac{128\beta\mu \max\{\mu_1, \mu\} r(r_a + r_b) \ln n}{3} \geq \frac{32\beta r\mu^2(r_a + r_b) \ln n}{3}, \\
\Omega_1 &= \frac{8\beta\mu^2(r_a r_b + r^2) \ln n}{3}.
\end{aligned}
$$

## 1.1 Deterministic Conditions for $Z_0$ to Be the Unique Minimizer

In this subsection, we will give the lemma stating two deterministic conditions for $Z_0$ to be the unique minimizer of (2),

**Lemma 1.** *We assume that there exists a matrix $Y \in \mathbb{R}^{n \times m}$ such that*

$$\textbf{A1} \quad \mathcal{R}_\Omega(Y) = Y, \ \|P_T(Y) - UV^\top\|_F \leq \sqrt{\frac{r}{2r_a}}, \ \|P_{T^\perp}(Y)\| < \frac{1}{2}.$$

*We further assume that for any $F \neq 0, F \in \mathbb{R}^{n \times m}$ satisfying $\mathcal{R}_\Omega(F) = 0$ and $F = P_A F P_B$, we have*

$$\textbf{A2} \quad \|P_T(F)\|_F \leq \gamma \|P_{T^\perp}(F)\|_F,$$

*where*

$$\gamma \leq \sqrt{\frac{r_a}{2r}}. \tag{5}$$

*Then, $Z_0$ is the unique minimizer to (2).*

*Proof.* We prove by contradiction. Assume there exists another solution $Z_0 + E$, with $E \neq 0$. We can further conclude that $AEB^T \neq 0$ (because if $AEB^T = 0$, then $A^T AEB^T B = IEI = 0$ for $A$ and $B$ are orthonormal, such that $E = 0$, leading to contradiction).

$Z_0 + E$ is a solution to (2) means that $\mathcal{R}_\Omega(A(Z_0 + E)B^T) = \mathcal{R}_\Omega(AZ_0B^T), \|Z_0 + E\|_{tr} \leq \|Z_0\|_{tr}$, and $A(Z_0 + E)B^T = P_A(A(Z_0 + E)B^T)P_B$. Evidently, we have $AEB^T = P_A(AEB^T)P_B$, $\mathcal{R}_\Omega(AEB^T) = 0$. Because $AEB^T \neq 0$, with Condition **A2**, we have $\|P_T(AEB^T)\|_F \leq \gamma\|P_{T^\perp}(AEB^T)\|_F \leq \gamma\|P_{T^\perp}(AEB^T)\|_{tr}$. We will use this fact later.

Let $U_\perp$ and $V_\perp$ be the left and right singular vectors of $P_{T^\perp}(AEB^T)$. Evidently, column vectors in $U_\perp$ are orthogonal to the column vectors in $U$, and column vectors in $V_\perp$ are orthogonal to the column vectors in $V$, that is $U^T U_\perp = 0$ and $V^T V_\perp = 0$. We have

$$\begin{aligned}
&\|Z_0 + E\|_{tr} \\
=\ & \|A(Z_0 + E)B^T\|_{tr} & (6) \\
=\ & \|A(Z_0 + E)B^T\|_{tr}\|UV^\top + U_\perp V_\perp^\top\| & (7) \\
\geq\ & \langle A(Z_0 + E)B^T, UV^\top + U_\perp V_\perp^\top\rangle & (8) \\
=\ & \langle AZ_0B^T, UV^\top\rangle + \langle AZ_0B^T, U_\perp V_\perp^\top\rangle + \langle AEB^T, UV^\top + U_\perp V_\perp^\top\rangle \\
=\ & \|M\|_{tr} + \langle AEB^T, -Y + UV^\top + U_\perp V_\perp^\top\rangle & (9) \\
=\ & \|M\|_{tr} + \langle AEB^T, UV^\top - P_T(Y) + U_\perp V_\perp^\top - P_{T^\perp}(Y)\rangle \\
=\ & \|M\|_{tr} + \langle P_T(AEB^T), UV^\top - P_T(Y)\rangle + \langle P_{T^\perp}(AEB^T), U_\perp V_\perp^\top - P_{T^\perp}(Y)\rangle \\
=\ & \|M\|_{tr} + \langle P_T(AEB^T), UV^\top - P_T(Y)\rangle + \langle P_{T^\perp}(AEB^T), U_\perp V_\perp^\top\rangle \\
& - \langle P_{T^\perp}(AEB^T), P_{T^\perp}(Y)\rangle \\
\geq\ & \|M\|_{tr} - \|P_T(AEB^T)\|_F\|UV^\top - P_T(Y)\|_F + \|P_{T^\perp}(AEB^T)\|_{tr} & (10) \\
& - \|P_{T^\perp}(Y)\|\|P_{T^\perp}(AEB^T)\|_{tr} \\
=\ & \|M\|_{tr} - \|P_T(AEB^T)\|_F\|UV^\top - P_T(Y)\|_F + (1 - \|P_{T^\perp}(Y)\|)\|P_{T^\perp}(AEB^T)\|_{tr} \\
>\ & \|M\|_{tr} - \sqrt{\frac{r}{2r_a}}\|P_T(AEB^T)\|_F + \frac{1}{2}\|P_{T^\perp}(AEB^T)\|_{tr} & (11) \\
\geq\ & \|M\|_{tr} + \|P_{T^\perp}(AEB^T)\|_{tr}\left(\frac{1}{2} - \gamma\sqrt{\frac{r}{2r_a}}\right) & (12) \\
=\ & \|Z_0\|_{tr} + \|P_{T^\perp}(AEB^T)\|_{tr}\left(\frac{1}{2} - \gamma\sqrt{\frac{r}{2r_a}}\right), & (13)
\end{aligned}$$

where

- (6) is because $A$ and $B$ are orthonormal;

- (7) is because $\|UV^\top + U_\perp V_\perp^\top\| = 1$;

- (8) is because $\langle M_1, M_2 \rangle \leq \|M_1\|\|M_2\|_{tr}$;

- (9) is because $\mathcal{R}_\Omega(Y) = Y$ and $\mathcal{R}_\Omega(AEB^T) = 0$ such that $\langle AEB^T, Y \rangle = 0$;

- (10) is because $\langle M_1, M_2 \rangle \leq \|M_1\|\|M_2\|_{tr}$, $\langle M_1, M_2 \rangle \leq \|M_1\|_F\|M_2\|_F$ and that $\|M\|_{tr} = <M, UV^T>$, where $U$ and $V$ are left and right sigular vectors of $M$;

- (11) is because of Condition **A1**;

- (12) is because of Condition **A2** and that Frobenius norm is smaller than trace norm;

- (13) is the same as (6).

When

$$\frac{1}{2} \geq \gamma\sqrt{\frac{r}{2r_a}},$$

that is,

$$\gamma \leq \sqrt{\frac{r_a}{2r}},$$

we have

$$\|Z_0 + E\|_{tr} > \|Z_0\|_{tr},$$

leading to the contradiction. $\qquad\square$

## 1.2 When will Condition A2 Hold with High Probability

In this section, we will give Lemma 7 stating when **A2** will hold with high probability.

### 1.2.1 Noncommutative Bernstein Inequality and its Derivations

First, we rewrite the Bernstein Inequality (Theorem 3.2 in [4]) and its derivations, which will be used later.

**Theorem 2.** *(Theorem 3.2 in [4]) Let* $\mathbf{X}_1, \dots, \mathbf{X}_L$ *be independent zero-mean random matrices of dimension* $d_1 \times d_2$. *Suppose* $\rho_k^2 = \max\{\|\mathbb{E}[\mathbf{X}_k\mathbf{X}_k^T]\|, \|\mathbb{E}[\mathbf{X}_k^T\mathbf{X}_k]\|\}$ *and* $\|\mathbf{X}_k\| \leq M$ *almost surely for all* $k$. *Then for any* $\tau > 0$,

$$\mathbb{P}\left[\left\|\sum_{k=1}^L \mathbf{X}_k\right\| > \tau\right] \leq (d_1 + d_2)\exp\left(\frac{-\tau^2/2}{\sum_{k=1}^L \rho_k^2 + M\tau/3}\right).$$

And we can easily have,

**Lemma 2.** *Let* $\mathbf{X}_1, \dots, \mathbf{X}_L$ *be independent zero-mean random matrices of dimension* $d_1 \times d_2$. *Suppose* $\max\{\|\mathbb{E}[\mathbf{X}_k\mathbf{X}_k^T]\|, \|\mathbb{E}[\mathbf{X}_k^T\mathbf{X}_k]\|\} \leq \rho_k^2$ *and* $\|\mathbf{X}_k\| \leq M$ *almost surely for all* $k$. *Then for any* $\tau > 0$,

$$\mathbb{P}\left[\left\|\sum_{k=1}^L \mathbf{X}_k\right\| > \tau\right] \leq (d_1 + d_2)\exp\left(\frac{-\tau^2/2}{\sum_{k=1}^L \rho_k^2 + M\tau/3}\right).$$

We then give a lemma derived from Lemma 2,

**Lemma 3.** *Let* $\mathbf{X}_1, \dots, \mathbf{X}_L$ *be independent zero-mean random matrices of dimension* $d_1 \times d_2$. *Suppose* $\max\{\|\mathbb{E}[\mathbf{X}_k\mathbf{X}_k^T]\|, \|\mathbb{E}[\mathbf{X}_k^T\mathbf{X}_k]\|\} \leq \rho_k^2$ *and* $\mathbf{X}_k \leq M$ *almost surely for all* $k$. *We assume that*

$$M^2 \ln\frac{d_1 + d_2}{\delta} \leq \frac{3}{8}\sum \rho_k^2,$$

*then with a probability at least* $1 - \delta$, *we have,*

$$\|\sum_{k=1}^L \mathbf{X}_k\| \leq \sqrt{\frac{8}{3}\ln\frac{d_1 + d_2}{\delta}\sum_{k=1}^L \rho_k^2}.$$

*Proof.* Assume $\tau = \sqrt{\frac{8}{3} \ln \frac{d_1 + d_2}{\delta} \sum_{k=1}^{L} \rho_k^2}$ (such that $\delta = (d_1 + d_2) \exp\left(\frac{-3\tau^2}{8 \sum \rho_k^2}\right)$), then we have,

$$
\begin{aligned}
M\tau &= \sqrt{M^2 \frac{8}{3} \ln \frac{d_1 + d_2}{\delta} \sum_{k=1}^{L} \rho_k^2} \\
&\leq \sqrt{\frac{3}{8} \sum \rho_k^2 \frac{8}{3} \sum_{k=1}^{L} \rho_k^2} \\
&= \sum_{k=1}^{L} \rho_k^2,
\end{aligned}
$$

such that

$$
(d_1 + d_2) \exp\left(\frac{-\tau^2/2}{\sum \rho_k^2 + M\tau/3}\right) \leq (d_1 + d_2) \exp\left(\frac{-3\tau^2}{8 \sum \rho_k^2}\right).
$$

Based on the Lemma 2, we have,

$$
\mathbb{P}\left[\| \sum_{k=1}^{L} \mathbf{X}_k \| > \tau \right] \leq (d_1 + d_2) \exp\left(\frac{-\tau^2/2}{\sum \rho_k^2 + M\tau/3}\right) \leq (d_1 + d_2) \exp\left(\frac{-3\tau^2}{8 \sum \rho_k^2}\right) = \delta,
$$

that is

$$
\mathbb{P}\left[\| \sum_{k=1}^{L} \mathbf{X}_k \| > \sqrt{\frac{8}{3} \ln \frac{d_1 + d_2}{\delta} \sum_{k=1}^{L} \rho_k^2} \right] \leq \delta.
$$

$\square$

We then give Lemma 4 which is also derived from Lemma 2,

**Lemma 4.** *Let* $\mathbf{X}_1, \ldots, \mathbf{X}_L$ *be independent zero-mean random matrices of dimension* $d_1 \times d_2$. *Suppose* $\max\{\|\mathbb{E}[\mathbf{X}_k \mathbf{X}_k^T]\|, \|\mathbb{E}[\mathbf{X}_k^T \mathbf{X}_k]\|\} \leq \rho_k^2$ *and* $\mathbf{X}_k \leq M$ *almost surely for all* $k$. *We assume that*

$$
M^2 \ln \frac{d_1 + d_2}{\delta} \geq \frac{3}{8} \sum \rho_k^2.
$$

*Then with a probability at least* $1 - \delta$, *we have,*

$$
\| \sum_{k=1}^{L} \mathbf{X}_k \| \leq \frac{8}{3} M \ln \frac{d_1 + d_2}{\delta}.
$$

*Proof.* Assume $\tau = \frac{8}{3} M \ln \frac{d_1 + d_2}{\delta}$ (such that $\delta = (d_1 + d_2) \exp\left(\frac{-3\tau}{8M}\right)$), then we have,

$$
M\tau = \frac{8}{3} M^2 \ln \frac{d_1 + d_2}{\delta} \geq \sum_{k=1}^{L} \rho_k^2,
$$

such that

$$
(d_1 + d_2) \exp\left(\frac{-\tau^2/2}{\sum \rho_k^2 + M\tau/3}\right) \leq (d_1 + d_2) \exp\left(\frac{-3\tau}{8M}\right) = \delta.
$$

Based on Lemma 2, we have

$$
\mathbb{P}\left[\| \sum_{k=1}^{L} \mathbf{X}_k \| > \tau \right] \leq (d_1 + d_2) \exp\left(\frac{-\tau^2/2}{\sum \rho_k^2 + M\tau/3}\right) \leq (d_1 + d_2) \exp\left(\frac{-3\tau}{8M}\right) = \delta,
$$

that is

$$
\mathbb{P}\left[\| \sum_{k=1}^{L} \mathbf{X}_k \| > \frac{8}{3} M \ln \frac{d_1 + d_2}{\delta} \right] \leq \delta.
$$

$\square$

### 1.2.2 Bounding $\|P_T - \frac{mn}{|\Omega|} P_T \mathcal{R}_\Omega P_T\|$

In this subsection, we will bound $\|P_T - \frac{mn}{|\Omega|} P_T \mathcal{R}_\Omega P_T\|$ in Lemma 5.

**Lemma 5.** *With a probability at least $1 - 2n^{-\beta+1}$, we have*

$$\left\| P_T - \frac{mn}{|\Omega|} P_T \mathcal{R}_\Omega P_T \right\| \leq \sqrt{\frac{8\beta \mu^2 r(r_a + r_b) \ln n}{3|\Omega|}},$$

*if*

$$|\Omega| \geq \frac{8\beta}{3} \mu^2 r(r_a + r_b) \ln n,$$

*and therefore, for any $F \in \mathbb{R}^{n \times m}$,*

$$\frac{mn}{|\Omega|} \langle F, P_T \mathcal{R}_\Omega P_T(F) \rangle \geq \frac{1}{2} \|P_T(F)\|_F^2,$$

*if $|\Omega| \geq \Omega_0$.*

*Proof.* For any $F \in \mathbb{R}^{n \times m}$, we have

$$P_T \mathcal{R}_\Omega P_T(F) = \sum_{(i,j) \in \Omega} \langle P_T(F), \mathbf{e}_i \mathbf{e}_j^\top \rangle P_T(\mathbf{e}_i \mathbf{e}_j^\top) = \sum_{(i,j) \in \Omega} \langle F, P_T(\mathbf{e}_i \mathbf{e}_j^\top) \rangle P_T(\mathbf{e}_i \mathbf{e}_j^\top).$$

For any $i \in [n]$ and $j \in [m]$, define linear operator $\mathbf{T}_{i,j}$ as

$$\mathbf{T}_{i,j}(F) = \langle F, P_T(\mathbf{e}_i \mathbf{e}_j^\top) \rangle P_T(\mathbf{e}_i \mathbf{e}_j^\top) = P_T \mathcal{R}_{(i,j)} P_T(F),$$

where $\mathcal{R}_{(i,j)}(F) = \mathbf{e}_i \mathbf{e}_j^\top F_{i,j}$. We write $P_T \mathcal{R}_\Omega P_T(F)$ as

$$P_T \mathcal{R}_\Omega P_T(F) = \sum_{(i,j) \in \Omega} P_T \mathcal{R}_{(i,j)} P_T(F) = \sum_{(i,j) \in \Omega} \mathbf{T}_{i,j}(F).$$

Evidently, we have

$$\frac{1}{|\Omega|} \mathbb{E}[P_T \mathcal{R}_\Omega P_T(F)] = \frac{1}{mn} P_T(F).$$

In this way, our objective, that is the spectral norm of $P_T - \frac{mn}{|\Omega|} P_T \mathcal{R}_\Omega P_T$, can be seen as the spectral norm of a sum of $|\Omega|$ independent zero-mean random variables, i.e. $\frac{1}{|\Omega|} P_T - \frac{mn}{|\Omega|} \mathbf{T}_{i,j}$, where we use Lemma 3. In this way, We need to compute $M$ and $\rho^2$ as

$$\begin{aligned}
&\|\frac{1}{|\Omega|} P_T - \frac{mn}{|\Omega|} \mathbf{T}_{i,j}\| \\
\leq\ & \max\{\|\frac{1}{|\Omega|} P_T\|, \|\frac{mn}{|\Omega|} \mathbf{T}_{i,j}\|\} \\
=\ & \max\{\|\frac{1}{|\Omega|} P_T\|, \frac{mn}{|\Omega|} \underset{\|F\|_F = 1}{\arg\max} \|\langle F, P_T(\mathbf{e}_i \mathbf{e}_j^\top) \rangle P_T(\mathbf{e}_i \mathbf{e}_j^\top)\|_F\} \\
=\ & \max\{\|\frac{1}{|\Omega|} P_T\|, \frac{mn}{|\Omega|} \underset{\|F\|_F = 1}{\arg\max} \langle F, P_T(\mathbf{e}_i \mathbf{e}_j^\top) \rangle \|P_T(\mathbf{e}_i \mathbf{e}_j^\top)\|_F\} \\
=\ & \max\{\|\frac{1}{|\Omega|} P_T\|, \frac{mn}{|\Omega|} \|P_T(\mathbf{e}_i \mathbf{e}_j^\top)\|_F^2\}.
\end{aligned}$$

Then our objective is to bound $\|P_T(\mathbf{e}_i\mathbf{e}_j^\top)\|_F^2$,

$$
\begin{aligned}
& \|P_T(\mathbf{e}_i\mathbf{e}_j^\top)\|_F^2 \\
=\ & \langle P_T(\mathbf{e}_i\mathbf{e}_j^\top), \mathbf{e}_i\mathbf{e}_j^\top \rangle \\
=\ & \langle P_A(\mathbf{e}_i\mathbf{e}_j^\top)P_V, \mathbf{e}_i\mathbf{e}_j^\top \rangle + \langle P_U(\mathbf{e}_i\mathbf{e}_j^\top)P_B, \mathbf{e}_i\mathbf{e}_j^\top \rangle - \langle P_U(\mathbf{e}_i\mathbf{e}_j^\top)P_V, \mathbf{e}_i\mathbf{e}_j^\top \rangle \\
=\ & \|P_A(\mathbf{e}_i\mathbf{e}_j^\top)P_V\|_F^2 + \|P_U(\mathbf{e}_i\mathbf{e}_j^\top)P_B\|_F^2 - \|P_U(\mathbf{e}_i\mathbf{e}_j^\top)P_V\|_F^2 \\
\leq\ & \|P_A\mathbf{e}_i\|_F^2\|P_V\mathbf{e}_j\|_F^2 + \|P_U\mathbf{e}_i\|_F^2\|P_B\mathbf{e}_j\|_F^2 \\
\leq\ & \frac{r_a\mu_{AB}}{n}\frac{r\mu_0}{m} + \frac{r\mu_0}{n}\frac{r_b\mu_{AB}}{m} \\
=\ & \frac{r\mu_0\mu_{AB}(r_a+r_b)}{mn} \leq \frac{r\mu^2(r_a+r_b)}{mn}.
\end{aligned}
$$

Thus

$$
\begin{aligned}
\|\frac{1}{|\Omega|}P_T - \frac{mn}{|\Omega|}\mathbf{T}_{i,j}\| &\leq \max\{\|\frac{1}{|\Omega|}P_T\|, \frac{mn}{|\Omega|}\|P_T(\mathbf{e}_i\mathbf{e}_j^\top)\|_F^2\} \\
&\leq \max\{\|\frac{1}{|\Omega|}P_T\|, \frac{r\mu^2(r_a+r_b)}{|\Omega|}\} \\
&= \max\{\frac{1}{|\Omega|}, \frac{r\mu^2(r_a+r_b)}{|\Omega|}\} \\
&= \frac{r\mu^2(r_a+r_b)}{|\Omega|} = M.
\end{aligned}
$$

Then we calculate $\rho_{i,j}^2$, that is

$$
\rho_{i,j}^2 = \max\{\|\mathbb{E}[(\frac{1}{|\Omega|}P_T - \frac{mn}{|\Omega|}\mathbf{T}_{i,j})^T(\frac{1}{|\Omega|}P_T - \frac{mn}{|\Omega|}\mathbf{T}_{i,j})]\|, \|\mathbb{E}[(\frac{1}{|\Omega|}P_T - \frac{mn}{|\Omega|}\mathbf{T}_{i,j})(\frac{1}{|\Omega|}P_T - \frac{mn}{|\Omega|}\mathbf{T}_{i,j})^T]\|\}.
$$

If we just consider one part of the $\max\{\}$, we have

$$
\begin{aligned}
\rho_{i,j}^2 &= \|\mathbb{E}[(\frac{1}{|\Omega|}P_T - \frac{mn}{|\Omega|}\mathbf{T}_{i,j})^T(\frac{1}{|\Omega|}P_T - \frac{mn}{|\Omega|}\mathbf{T}_{i,j})]\| \\
&= \|\mathbb{E}[\frac{1}{|\Omega|^2}P_TP_T + \frac{m^2n^2}{|\Omega|^2}\mathbf{T}_{i,j}\mathbf{T}_{i,j} - \frac{2mn}{|\Omega|^2}P_T\mathbf{T}_{i,j}]\| \\
&= \|\mathbb{E}[\frac{1}{|\Omega|^2}P_TP_T + \frac{m^2n^2}{|\Omega|^2}\mathbf{T}_{i,j}\mathbf{T}_{i,j} - \frac{2mn}{|\Omega|^2}P_T\mathbf{T}_{i,j}]\| \\
&= \|\frac{1}{|\Omega|^2}P_T + \frac{m^2n^2}{|\Omega|^2}\mathbb{E}[\mathbf{T}_{i,j}\mathbf{T}_{i,j}] - \frac{2mn}{|\Omega|^2}P_T\mathbb{E}[\mathbf{T}_{i,j}]\| \\
&= \|\frac{1}{|\Omega|^2}P_T + \frac{m^2n^2}{|\Omega|^2}\mathbb{E}[\mathbf{T}_{i,j}\mathbf{T}_{i,j}] - \frac{2mn}{|\Omega|^2}P_T\frac{1}{mn}P_T\| \\
&= \|\frac{m^2n^2}{|\Omega|^2}\mathbb{E}[\mathbf{T}_{i,j}\mathbf{T}_{i,j}] - \frac{1}{|\Omega|^2}P_T\| \leq \max\{\frac{m^2n^2}{|\Omega|^2}\mathbb{E}[\|\mathbf{T}_{i,j}\mathbf{T}_{i,j}\|], \frac{1}{|\Omega|^2}\} \\
&= \max\{\frac{m^2n^2}{|\Omega|^2}\mathbb{E}[\|P_T(\mathbf{e}_i\mathbf{e}_j^\top)\|_F^2\|\mathbf{T}_{i,j}\|], \frac{1}{|\Omega|^2}\} \\
&\leq \max\{\frac{m^2n^2}{|\Omega|^2}\max(\|P_T(\mathbf{e}_i\mathbf{e}_j^\top)\|_F^2)\mathbb{E}[\|\mathbf{T}_{i,j}\|], \frac{1}{|\Omega|^2}\} \\
&\leq \max\{\frac{m^2n^2}{|\Omega|^2}\frac{r\mu^2(r_a+r_b)}{mn}\frac{1}{mn}\|P_T\|, \frac{1}{|\Omega|^2}\} \\
&= \frac{r\mu^2(r_a+r_b)}{|\Omega|^2}.
\end{aligned}
$$

Since Lemma 3, using $M = \frac{r\mu^2(r_a+r_b)}{|\Omega|}$ and $\rho^2 = \frac{r\mu^2(r_a+r_b)}{|\Omega|^2}$, we have, with a probability $1 - 2n^{-\beta+1}$,

$$\left\| P_T - \frac{mn}{|\Omega|} P_T \mathcal{R}_\Omega P_T \right\| \leq \sqrt{\frac{8}{3} \ln \frac{n+m}{2n^{-\beta+1}} \frac{r\mu^2(r_a+r_b)}{|\Omega|}}$$

$$\leq \sqrt{\frac{8\beta r\mu^2(r_a+r_b)\ln n}{3|\Omega|}},$$

with the condition

$$\frac{r^2\mu^4(r_a+r_b)^2}{|\Omega|^2} \ln \frac{n+m}{2n^{-\beta+1}} \leq \frac{3}{8} \frac{r\mu^2(r_a+r_b)}{|\Omega|},$$

that is

$$\ln \frac{n+m}{2n^{-\beta+1}} \leq \frac{3|\Omega|}{8r\mu^2(r_a+r_b)}.$$

we can tight the condition to,

$$\ln \frac{2n}{2n^{-\beta+1}} \leq \frac{3|\Omega|}{8r\mu^2(r_a+r_b)},$$

that is

$$|\Omega| \geq \frac{8\beta r\mu^2(r_a+r_b)\ln n}{3}.$$

If

$$|\Omega| \geq \Omega_0 \geq \frac{32\beta r\mu^2(r_a+r_b)\ln n}{3},$$

that is to say,

$$\left\| P_T - \frac{mn}{|\Omega|} P_T \mathcal{R}_\Omega P_T \right\| \leq \sqrt{\frac{8\beta r\mu^2(r_a+r_b)\ln n}{3|\Omega|}} \leq \frac{1}{2},$$

we can have, following the property of matrix norm, that

$$\langle F, P_T(F) - \frac{mn}{|\Omega|} P_T \mathcal{R}_\Omega P_T(F) \rangle \leq \frac{1}{2} \|P_T(F)\|_F^2,$$

from which we will have,

$$\langle F, P_T(F) \rangle - \langle F, \frac{mn}{|\Omega|} P_T \mathcal{R}_\Omega P_T(F) \rangle \leq \frac{1}{2} \|P_T(F)\|_F^2.$$

from which we will further have,

$$\frac{1}{2} \|P_T(F)\|_F^2 \leq \frac{mn}{|\Omega|} \langle F, P_T \mathcal{R}_\Omega P_T(F) \rangle.$$

$\square$

### 1.2.3 Bounding $\|P_{T^\perp} - \frac{mn}{|\Omega|} P_{T^\perp} \mathcal{R}_\Omega P_{T^\perp}\|$

We will give the result of bounding $\|P_{T^\perp} - \frac{mn}{|\Omega|} P_{T^\perp} \mathcal{R}_\Omega P_{T^\perp}\|$ in Lemma 6.

**Lemma 6.** *With a probability at least $1 - 2n^{-\beta+1}$, we have, if $|\Omega| \leq \frac{8\beta\mu^2(r_a r_b + r^2)\ln n}{3} = \Omega_1$, then*

$$\left\| P_{T^\perp} - \frac{mn}{|\Omega|} P_{T^\perp} \mathcal{R}_\Omega P_{T^\perp} \right\| \leq \frac{8\beta\mu^2(r_a r_b + r^2)\ln n}{3|\Omega|},$$

*and thus for any $Z \in \mathbb{R}^{m\times n}$,*

$$\frac{mn}{|\Omega|} \langle Z, P_{T^\perp} \mathcal{R}_\Omega P_{T^\perp}(Z) \rangle \leq \frac{16\beta\mu^2(r_a r_b + r^2)\ln n}{3|\Omega|} \|P_{T^\perp}(Z)\|_F^2.$$

*Proof.* Similar to the proof of Lemma 5, we have

$$P_{T^\perp} \mathcal{R}_\Omega P_{T^\perp}(F) \quad = \quad \sum_{(i,,j)\in\Omega} P_{T^\perp} \mathcal{R}_{(i,j)} P_{T^\perp}(F) = \sum_{(i,j)\in\Omega} \mathbf{T}_{i,j}(F),$$

where $\mathbf{T}_{i,j} = P_{T^\perp} \mathcal{R}_{(i,j)} P_{T^\perp}$. Evidently, we have

$$\mathrm{E}[P_{T^\perp} \mathcal{R}_\Omega P_{T^\perp}] = \frac{|\Omega|}{mn} P_{T^\perp}.$$

Now our objective, that is $P_{T^\perp} - \frac{mn}{|\Omega|} P_{T^\perp} \mathcal{R}_\Omega P_{T^\perp}$, can be seen as a sum of $|\Omega|$ independent zero-mean variables, i.e., $\frac{1}{|\Omega|} P_{T^\perp} - \frac{mn}{|\Omega|} \mathbf{T}_{i,j}$. In this way, we compute $M$ and $\rho^2$ as,

$$
\begin{aligned}
\|\frac{1}{|\Omega|} P_{T^\perp} - \frac{mn}{|\Omega|} \mathbf{T}_{i,j}\| \quad &\leq \quad \max\{\frac{1}{|\Omega|}\|P_{T^\perp}\|, \frac{mn}{|\Omega|}\|\mathbf{T}_{i,j}\|\} \\
&\leq \quad \max\{\frac{1}{|\Omega|}\|P_{T^\perp}\|, \frac{mn}{|\Omega|}\|P_{T^\perp}(\mathbf{e}_i\mathbf{e}_j^T)\|_F^2\},
\end{aligned}
$$

and

$$
\begin{aligned}
\|P_{T^\perp}(\mathbf{e}_i\mathbf{e}_j^T)\|_F^2 \quad &= \quad \|P_A(\mathbf{e}_i\mathbf{e}_j^T)P_B\|_F^2 + \|P_U(\mathbf{e}_i\mathbf{e}_j^T)P_V\|_F^2 - \|P_A(\mathbf{e}_i\mathbf{e}_j^T)P_V\|_F^2 - \|P_U(\mathbf{e}_i\mathbf{e}_j^T)P_B\|_F^2 \\
&\leq \quad \|P_A(\mathbf{e}_i\mathbf{e}_j^T)P_B\|_F^2 + \|P_U(\mathbf{e}_i\mathbf{e}_j^T)P_V\|_F^2 \\
&\leq \quad \frac{\mu_{AB}r_a}{n}\frac{\mu_{AB}r_b}{m} + \frac{\mu_0 r}{n}\frac{\mu_0 r}{m} \\
&= \quad \frac{\mu_{AB}^2 r_a r_b + \mu_0^2 r^2}{mn} \\
&\leq \quad \frac{\mu^2(r_a r_b + r^2)}{mn},
\end{aligned}
$$

such that

$$M = \max\{\frac{1}{|\Omega|}, \frac{mn}{|\Omega|}\frac{\mu^2(r_a r_b + r^2)}{mn}\} = \frac{\mu^2(r_a r_b + r^2)}{|\Omega|}.$$

In the same way,

$$
\begin{aligned}
\rho^2 \quad &= \quad \|\mathbb{E}[(\frac{1}{|\Omega|}P_{T^\perp} - \frac{mn}{|\Omega|}\mathbf{T}_{i,j})^T(\frac{1}{|\Omega|}P_{T^\perp} - \frac{mn}{|\Omega|}\mathbf{T}_{i,j})]\| \\
&= \quad \|\frac{m^2n^2}{|\Omega|^2}\mathbb{E}[\mathbf{T}_{i,j}\mathbf{T}_{i,j}] - \frac{1}{|\Omega|^2}P_{T^\perp}\| \\
&\leq \quad \max\{\frac{m^2n^2}{|\Omega|^2}\frac{\mu^2(r_a r_b + r^2)}{mn}\frac{1}{mn}, \frac{1}{|\Omega|^2}\} \\
&= \quad \frac{\mu^2(r_a r_b + r^2)}{|\Omega|^2}.
\end{aligned}
$$

Using Lemma 3, we have, with probability at least $1 - 2n^{-\beta+1}$, we have, if

$$\frac{\mu^4(r_a r_b + r^2)^2}{|\Omega|^2}\ln\frac{2n}{2n^{-\beta+1}} \geq \frac{3}{8}\frac{\mu^2(r_a r_b + r^2)}{|\Omega|},$$

that is

$$|\Omega| \leq \frac{8\beta\mu^2(r_a r_b + r^2)\ln n}{3} = \Omega_1,$$

we have,

$$
\begin{aligned}
\left\|P_{T^\perp} - \frac{mn}{|\Omega|}P_{T^\perp}\mathcal{R}_\Omega P_{T^\perp}\right\| \quad &\leq \quad \frac{8}{3}\frac{\mu^2(r_a r_b + r^2)}{|\Omega|}\ln\frac{2n}{2n^{-\beta+1}} \\
&= \quad \frac{8\beta\mu^2(r_a r_b + r^2)\ln n}{3|\Omega|}.
\end{aligned}
$$

And due to the property of matrix norm, we further have,

$$
\begin{aligned}
-\frac{8\beta\mu^2(r_ar_b+r^2)\ln n}{3|\Omega|}\|P_{T^\perp}(F)\|_F^2 &\leq \langle F, P_{T^\perp}(F) - \frac{mn}{|\Omega|}P_{T^\perp}\mathcal{R}_\Omega P_{T^\perp}(F)\rangle \\
&= \langle F, P_{T^\perp}(F)\rangle - \langle F, \frac{mn}{|\Omega|}P_{T^\perp}\mathcal{R}_\Omega P_{T^\perp}(F)\rangle \\
&= \|P_{T^\perp}(F)\|_F^2 - \langle F, \frac{mn}{|\Omega|}P_{T^\perp}\mathcal{R}_\Omega P_{T^\perp}(F)\rangle,
\end{aligned}
$$

thus

$$
\langle F, \frac{mn}{|\Omega|}P_{T^\perp}\mathcal{R}_\Omega P_{T^\perp}(F)\rangle \leq (1 + \frac{8\beta\mu^2(r_ar_b+r^2)\ln n}{3|\Omega|})\|P_{T^\perp}(F)\|_F^2.
$$

Because of

$$
|\Omega| \leq \frac{8\beta\mu^2(r_ar_b+r^2)\ln n}{3} = \Omega_1,
$$

we can have

$$
(1 + \frac{8\beta\mu^2(r_ar_b+r^2)\ln n}{3|\Omega|}) \leq \frac{16\beta\mu^2(r_ar_b+r^2)\ln n}{3|\Omega|},
$$

thus

$$
\langle F, \frac{mn}{|\Omega|}P_{T^\perp}\mathcal{R}_\Omega P_{T^\perp}(F)\rangle \leq \frac{16\beta\mu^2(r_ar_b+r^2)\ln n}{3|\Omega|}\|P_{T^\perp}(F)\|_F^2.
$$

$\square$

### 1.2.4 Proof of A2 Holding with High Probability

Based on Lemma 5 and 6, we can give the result stating when **A2** will hold with high probability,

**Lemma 7.** *With a probability $1 - 4n^{-\beta+1}$, for any $F \neq 0, F \in \mathbb{R}^{n\times m}$ satisfying $\mathcal{R}_\Omega(F) = 0$ and $F = P_A F P_B$, we have*

$$
\|P_T(F)\|_F \leq \gamma\|P_{T^\perp}(F)\|_F,
$$

*where $\gamma$ is given in (5), provided*

$$
\Omega_0 \leq |\Omega| \leq \Omega_1.
$$

*Proof.* Since $\mathcal{R}_\Omega(F) = 0$ and $F = P_A F P_B$, we have $\mathcal{R}_\Omega P_T(F) = -\mathcal{R}_\Omega P_{T^\perp}(F)$. Thus we have

$$
\frac{mn}{|\Omega|}\langle F, P_T\mathcal{R}_\Omega P_T(F)\rangle = \frac{mn}{|\Omega|}\langle F, P_{T^\perp}\mathcal{R}_\Omega P_{T^\perp}(F)\rangle.
$$

First, according to Lemma 5, and Lemma 6, with a probability at least $1 - 4n^{-\beta+1}$, we have

$$
\begin{aligned}
\frac{1}{2}\|P_T(F)\|_F^2 \leq \frac{mn}{|\Omega|}\langle F, P_T\mathcal{R}_\Omega P_T(F)\rangle &= \frac{mn}{|\Omega|}\langle F, P_{T^\perp}\mathcal{R}_\Omega P_{T^\perp}(F)\rangle \\
&\leq \frac{16\beta\mu^2(r_ar_b+r^2)\ln n}{3|\Omega|}\|P_{T^\perp}(Z)\|_F^2 \\
&\leq \frac{16\beta\mu^2(r_ar_b+r^2)\ln n}{3\Omega_0}\|P_{T^\perp}(Z)\|_F^2 \\
&= \frac{1}{2}\|P_{T^\perp}(Z)\|_F^2,
\end{aligned}
$$

such that

$$
\|P_T(F)\|_F \leq \frac{1}{\sqrt{2}}\|P_{T^\perp}(Z)\|_F.
$$

When $r \ll r_a$, surely we have $\frac{1}{\sqrt{2}} \leq \sqrt{\frac{r_a}{2r}}$, thus completing our proof. $\square$

## 1.3 When will Condition A1 Hold with High Probability

Before showing the result when will condition **A1** hold with high probability, we will bound the following two values $\frac{mn}{|\Omega|}\|P_{T^\perp}\mathcal{R}_\Omega P_T(F)\|$ and $\|P_T(F) - \frac{mn}{|\Omega|}P_T\mathcal{R}_\Omega P_T(F)\|_\infty$ where $\|\cdot\|_\infty$ is the maximum entry of a matrix, in Lemma 8 and 9 respectively.

**Lemma 8.** *For a fixed $F \in \mathbb{R}^{n\times m}$, with a probability $1 - 2n^{-\beta+1}$, we have,*

$$\frac{mn}{|\Omega|}\|P_{T^\perp}\mathcal{R}_\Omega P_T(F)\| \le \|P_T(F)\|_\infty \sqrt{\frac{8\beta mn\mu r_a \ln n}{3|\Omega|}},$$

*if $|\Omega| \ge \Omega_0$.*

*Proof.* Similar to the proof for Lemma 5, we write

$$P_{T^\perp}\mathcal{R}_\Omega P_T(F) \quad = \quad \sum_{(i,j)\in\Omega}\langle F, P_T(\mathbf{e}_i\mathbf{e}_j^\top)\rangle P_{T^\perp}(\mathbf{e}_i\mathbf{e}_j^\top) = \sum_{(i,j)\in\Omega}\mathbf{T}_{i,j},$$

where

$$\mathbf{T}_{i,j}(F) = \langle F, P_T(\mathbf{e}_i\mathbf{e}_j^\top)\rangle P_{T^\perp}(\mathbf{e}_i\mathbf{e}_j^\top).$$

Evidently,

$$\mathrm{E}[P_{T^\perp}\mathcal{R}_\Omega P_T(F)] = 0.$$

To use Lemma 3, we compute $M$ and $\rho^2$ as,

$$
\begin{aligned}
M &= \max_{i\in[n],j\in[m]}\|\mathbf{T}_{i,j}\|\\
&\le \max_{i\in[n],j\in[m]}\max_{\|F\|_F=1}\|\langle F, P_T(\mathbf{e}_i\mathbf{e}_j^\top)\rangle P_{T^\perp}(\mathbf{e}_i\mathbf{e}_j^\top)\|_F\\
&\le \max_{i\in[n],j\in[m]}\langle F, P_T(\mathbf{e}_i\mathbf{e}_j^\top)\rangle\|P_{T^\perp}(\mathbf{e}_i\mathbf{e}_j^\top)\|\\
&\le \max_{i\in[n],j\in[m]}\mathcal{R}_{i,j}P_T(F)\|P_{T^\perp}(\mathbf{e}_i\mathbf{e}_j^\top)\|\\
&\le \|P_T(F)\|_\infty \max_{i\in[n],j\in[m]}\|P_{T^\perp}(\mathbf{e}_i\mathbf{e}_j)\|\\
&\le \|P_T(F)\|_\infty\sqrt{\frac{\mu^2(r_a r_b + r^2)}{mn}},
\end{aligned}
$$

and

$$
\begin{aligned}
\rho_{i,j}^2 &= \max\left\{\|\mathrm{E}[\mathbf{T}_{i,j}\mathbf{T}_{i,j}^*]\|, \|\mathrm{E}[\mathbf{T}_{i,j}^*\mathbf{T}_{i,j}]\|\right\}\\
&= \|P_T(F)\|_\infty^2\max\left\{\left\|\mathrm{E}\left[[P_{T^\perp}(\mathbf{e}_i\mathbf{e}_j^\top)]^\top[P_{T^\perp}(\mathbf{e}_i\mathbf{e}_j^\top)]\right]\right\|, \left\|\mathrm{E}\left[[P_{T^\perp}(\mathbf{e}_i\mathbf{e}_j^\top)][P_{T^\perp}(\mathbf{e}_i\mathbf{e}_j^\top)]^\top\right]\right\|\right\}\\
&= \|P_T(F)\|_\infty^2\max\left(\|\mathrm{E}[P_{B^\perp}\mathbf{e}_j\mathbf{e}_i^\top P_{A^\perp}\mathbf{e}_i\mathbf{e}_j^\top P_{B^\perp}]\|, \|\mathrm{E}[P_{A^\perp}\mathbf{e}_i\mathbf{e}_j^\top P_{B^\perp}\mathbf{e}_j\mathbf{e}_i^\top P_{A^\perp}]\|\right)\\
&\le \|P_T(F)\|_\infty^2\max\left(\frac{\mu_{AB}r_a}{n}\|\mathrm{E}[P_{B^\perp}\mathbf{e}_j\mathbf{e}_j^\top P_{B^\perp}]\|, \frac{\mu_{AB}r_b}{m}\|\mathrm{E}[P_{A^\perp}\mathbf{e}_i\mathbf{e}_i^\top P_{A^\perp}]\|\right)\\
&\le \|P_T(F)\|_\infty^2\max\left(\frac{\mu_{AB}r_a}{n}\|P_{B^\perp}\mathrm{E}[\mathbf{e}_j\mathbf{e}_j^\top]P_{B^\perp}\|, \frac{\mu_{AB}r_b}{m}\|P_{A^\perp}\mathrm{E}[\mathbf{e}_i\mathbf{e}_i^\top]P_{A^\perp}\|\right)\\
&\le \|P_T(F)\|_\infty^2\max\left(\frac{\mu_{AB}r_a}{mn}\|P_{B^\perp}P_{B^\perp}\|, \frac{\mu_{AB}r_b}{nm}\|P_{A^\perp}P_{A^\perp}\|\right)\\
&\le \|P_T(F)\|_\infty^2\frac{\mu_{AB}\max\{r_a, r_b\}}{mn}.
\end{aligned}
$$

Without loss of generality, we assume $r_b \le r_a$, such that we have,

$$\rho_{i,j}^2 \quad \le \quad \|P_T(F)\|_\infty^2\frac{\mu_{AB}r_a}{mn} \le \|P_T(F)\|_\infty^2\frac{\mu r_a}{mn}.$$

Use Lemma 3, we have, if

$$\|P_T(F)\|_\infty^2\frac{\mu^2(r_a r_b + r^2)}{mn}\ln\frac{2n}{2n^{-\beta+1}} \le \frac{3}{8}\|P_T(F)\|_\infty^2\frac{\mu r_a|\Omega|}{mn},$$

that is

$$\frac{8\mu(r_a r_b + r^2)\beta \ln n}{3r_a} \leq |\Omega|, \tag{14}$$

we have, with a probability $1 - 2n^{-\beta+1}$,

$$
\begin{aligned}
\frac{mn}{|\Omega|}\|P_{T^\perp}\mathcal{R}_\Omega P_T(F)\| &\leq \frac{mn}{|\Omega|}\|P_T(F)\|_\infty\sqrt{\frac{8\beta\rho^2|\Omega|\ln n}{3}} \\
&\leq \frac{mn}{|\Omega|}\|P_T(F)\|_\infty\sqrt{\frac{8\beta\mu r_a|\Omega|\ln n}{3mn}} \\
&= \|P_T(F)\|_\infty\sqrt{\frac{8\beta mn\mu r_a \ln n}{3|\Omega|}}.
\end{aligned}
$$

To prove (14) holds, we will using the condition $|\Omega| \geq \Omega_0$ (in Lemma 7). More specifically, we have,

$$|\Omega| \geq \Omega_0 \geq \frac{32\beta r\mu^2(r_a + r_b)\ln n}{3} \geq \frac{8\mu(r_a r_b + r^2)\beta \ln n}{3r_a},$$

where the third inequality is because that $4\mu(r_a + r_b) \geq r_b + r^2/r_a$, which is true because $\mu \geq 1$, $r_a \geq r_b$ and $r_a \gg r$.

Thus under the condition $|\Omega| \geq \Omega_0$, we complete our proof. $\qquad\square$

**Lemma 9.** *For a fixed $Z \in \mathbb{R}^{n\times m}$, with a probability $1 - 2n^{-\beta+2}$, we have*

$$\left\|\left(P_T - \frac{mn}{|\Omega|}P_T\mathcal{R}_\Omega P_T\right)(F)\right\|_\infty \leq \sqrt{\frac{8\beta r\mu^2(r_a + r_b)\ln n}{3|\Omega|}}\|P_T(F)\|_\infty,$$

*and therefore*

$$\left\|\left(P_T - \frac{mn}{|\Omega|}P_T\mathcal{R}_\Omega P_T\right)(F)\right\|_\infty \leq \frac{1}{2}\|P_T(F)\|_\infty,$$

*if $|\Omega| \geq \Omega_0$.*

*Proof.* This lemma can be proved by the standard Bernstein Inequality. For each matrix index $(a, b)$, sample $(i, j)$ uniformly at random to define the random variable

$$\xi_{a,b} = [mnP_T\mathcal{R}_{i,j}P_T(F) - P_T(F)]_{a,b}.$$

We have

$$\mathbb{E}[\xi_{a,b}] = 0,$$

$$|\xi_{a,b}| \leq \|P_T\mathcal{R}_{i,j}P_T - P_T\|\|P_T(F)\|_\infty \leq r\mu^2(r_a + r_b)\|P_T(F)\|_\infty,$$

and

$$
\begin{aligned}
\mathbb{E}[\xi_{a,b}^2] &= \mathbb{E}[([mnP_T\mathcal{R}_{i,j}P_T(F) - P_T(F)]_{a,b})^2] \\
&= \mathbb{E}[[m^2n^2P_T\mathcal{R}_{i,j}P_T(F)]_{a,b}^2] + [P_T(F)]_{a,b}^2 - 2mn\mathbb{E}[[P_T(F)]_{a,b}[P_T\mathcal{R}_{i,j}P_T(F)]_{a,b}] \\
&= m^2n^2\mathbb{E}[[P_T\mathcal{R}_{i,j}P_T(F)]_{a,b}^2] - [P_T(F)]_{a,b}^2 \\
&= m^2n^2\mathbb{E}\left[\left(\langle \mathbf{e}_a\mathbf{e}_b^\top, P_T(\mathbf{e}_i\mathbf{e}_j^\top)\rangle\langle F, P_T(\mathbf{e}_i\mathbf{e}_j^\top)\rangle\right)^2\right] - [P_T(F)]_{a,b}^2 \\
&= mn\|P_T(F)\|_F^2\|P_T(\mathbf{e}_a\mathbf{e}_b)\|_F^2 - [P_T(F)]_{a,b}^2 \\
&\leq \|P_T(F)\|_\infty^2 r\mu^2(r_a + r_b).
\end{aligned}
$$

Using the standard Bernstein's inequality, we have,

$$\mathbb{P}\left[|[mnP_T\mathcal{R}_\Omega P_T(F) - |\Omega|P_T(F)]_{a,b}| > \sqrt{\frac{8|\Omega|\|P_T(F)\|_\infty^2 r\mu^2(r_a+r_b)\ln\frac{2}{2n^{-\beta}}}{3}}\right] \le 2n^{-\beta},$$

that is,

$$\mathbb{P}\left[\left|\left[\frac{mn}{|\Omega|}P_T\mathcal{R}_\Omega P_T(F) - P_T(F)\right]_{a,b}\right| > \sqrt{\frac{8r\beta\mu^2(r_a+r_b)\ln n}{3|\Omega|}}\|P_T(F)\|_\infty\right] \le 2n^{-\beta},$$

if

$$(r\mu^2(r_a+r_b)\|P_T(F)\|_\infty)^2\beta\ln n \le \frac{3}{8}|\Omega|\|P_T(F)\|_\infty^2 r\mu^2(r_a+r_b),$$

which is,

$$\frac{(8r\mu^2(r_a+r_b))\beta\ln n}{3} \le |\Omega|.$$

Take the union bound, we have, with a probability at least $1 - 2n^{-\beta+2}$

$$\left\|\frac{mn}{|\Omega|}P_T\mathcal{R}_\Omega P_T(F) - P_T(F)\right\|_\infty \le \sqrt{\frac{8r\beta\mu^2(r_a+r_b)\ln n}{3|\Omega|}}\|P_T(F)\|_\infty.$$

If $|\Omega| \ge \Omega_0$, we have,

$$\left\|\frac{mn}{|\Omega|}P_T\mathcal{R}_\Omega P_T(F) - P_T(F)\right\|_\infty \le \frac{1}{2}\|P_T(F)\|_\infty.$$

$\square$

To verify there exists a matrix $Y$ that satisfies the conditions in **A1**, we follow the idea in [4] and construct $Y$ as follows. We randomly select $q\Omega_0$ entries from $\Omega$, where the value of $q$ will be discussed later, and divide the set of selected entries into $q$ subsets, denoted by $\Omega_1, \ldots, \Omega_q$, with

$$|\Omega_i| = \Omega_0, \quad i = 1, \ldots, q.$$

We generate a sequence of $Y_t, t = 1, \ldots, q$ as follows

$$Y_t = \frac{mn}{\Omega_0}\sum_{i=1}^t \mathcal{R}_{\Omega_i}(W_i),$$

where $W_1 = UV^\top$ and $W_{t+1}$ is defined inductively as,

$$\begin{aligned}
W_{t+1} = P_T(UV^\top - Y_t) &= W_t - \frac{mn}{\Omega_0}P_T\mathcal{R}_{\Omega_t}(W_t) \\
&= \left(P_T - \frac{mn}{\Omega_0}P_T\mathcal{R}_{\Omega_t}P_T\right)W_t \text{ (This is because } P_T(W_t) = W_t\text{)}.
\end{aligned}$$

We construct $Y$ as the last element of the sequence, i.e. $Y = Y_q$. Evidently, we have,

$$Y = \mathcal{R}_\Omega(Y). \tag{15}$$

The following two lemmas show that $Y$ satisfies the other two properties in assumption **A1**.

**Lemma 10.** *With a probability $1 - 2qn^{-\beta+1}$, we have*

$$\|P_T(Y) - UV^\top\| \le \sqrt{\frac{r}{2r_a}},$$

*if $q \ge q_0$.*

*Proof.* Since
$$W_{t+1} = (P_T - \frac{mn}{\Omega_0} P_T \mathcal{R}_{\Omega_t} P_T) W_t,$$
we have
$$\|P_T(Y) - UV^\top\| \leq \Pi_{i=1}^q \left\| P_T - \frac{mn}{\Omega_0} P_T \mathcal{R}_{\Omega_i} P_T \right\|.$$
Using Lemma 5, we have, with a probability $1 - 2qn^{-\beta+1}$,
$$\|P_T(Y) - UV^\top\| \leq \frac{1}{2^q},$$
and by choosing $q = q_0 = \frac{1}{2}(1 + \log_2 r_a - \log_2 r)$, we have $\|P_T(Y) - UV^\top\| \leq \sqrt{r/2r_a}$. $\qquad\square$

**Lemma 11.** *With a probability $1 - 2qn^{-\beta+1} - 2qn^{-\beta+2}$, we have*
$$\|P_{T^\perp}(Y)\| \leq \frac{1}{2}.$$

*Proof.* Because of Lemma 9 and $W_{t+1} = (P_T - \frac{mn}{\Omega_0} P_T \mathcal{R}_{\Omega_t} P_T) W_t$, we have
$$\|W_{t+1}\|_\infty = \|(P_T - \frac{mn}{\Omega_0} P_T \mathcal{R}_{\Omega_t} P_T) W_t\|_\infty \leq \frac{1}{2} \|W_t\|_\infty.$$
To bound $\|P_{T^\perp}(Y)\|$, we have,
$$
\begin{aligned}
\|P_{T^\perp}(Y)\| &\leq \sum_{i=1}^q \frac{mn}{\Omega_0} \|P_{T^\perp} \mathcal{R}_{\Omega_i} P_T(W_i)\| \\
&\leq \alpha \sum_{i=1}^q \|W_i\|_\infty \quad \text{(Lemma 8)} \\
&\leq \alpha \|W_1\|_\infty \sum_{i=1}^q \frac{1}{2^{i-1}} \\
&= 2\alpha \|W_1\|_\infty \\
&\leq 2 \times \sqrt{\frac{8\beta mn \mu r_a \ln n}{3|\Omega|}} \times \sqrt{\frac{u_1 r}{mn}} \\
&\leq 2 \times \sqrt{\frac{8\mu_1 r \beta \mu r_a \ln n}{3|\Omega|}},
\end{aligned}
$$
such that to bound $\|P_{T^\perp}(Y)\| \leq \frac{1}{2}$ we need,
$$|\Omega| \geq \frac{128\mu_1 r \beta \mu r_a \ln n}{3},$$
which is surely true when
$$|\Omega| \geq \frac{128\beta\mu \max\{\mu_1, \mu\} r(r_a + r_b) \ln n}{3} = \Omega_0.$$
$\qquad\square$

## 1.4 Proof of Theorem 1

*Proof.* Through Eq. 15, Lemma 10 and 11, condition **A1** in Lemma 1 is satisfied. Through Lemma 7, condition **A2** in Lemma 1 is satisfied. By the conclusion of Lemma 1 and union bound, we finish the proof. $\qquad\square$

Notice our condition here requires
$$q_0 \Omega_0 \leq \Omega_1,$$
that is
$$(1 + \log_2 r_a - \log_2 r)(8r(r_a + r_b)) \leq (r_a r_b + r^2),$$
which holds when $r_a \gg r$ and $r_b \gg a$.

**Algorithm 1** Maxide (Matrix Completion with Side Information)

---
1: **Initialization:** $\theta_1 = \theta_2 \in (0,1]$, $Z_1 = Z_2$, $L$, $\gamma > 1$, and stopping criterion $\epsilon$
2: $k = 2$;
3: **while** $\mathcal{L}(Z_{k+1}) \leq (1-\epsilon)\mathcal{L}(Z_k)$ **do**
4:     $Y_k = Z_k + \theta_k(\theta_{k-1}^{-1} - 1)(Z_k - Z_{k-1})$
5:     $Z_{k+1} = \arg\min_Z \lambda\|Z\|_{tr} + Q_k(Z)$
6:     **while** $\mathcal{E}(Z_{k+1}) - \mathcal{E}(Y_k) \geq H_k(L)$ **do**
7:        $L = L * \gamma$
8:        $Z_{k+1} = \arg\min_Z \lambda\|Z\|_{tr} + Q_k(Z)$
9:     **end while**
10:    $\theta_{k+1} = (\sqrt{\theta_k^4 + 4\theta_k^2} - \theta_k^2)/2$
11:    $k = k + 1$
12: **end while**

---

## 2 The Maxide Algorithm

Algorithm 1 gives the key steps for solving the optimization problem in (3), where $\mathcal{E}(Z)$, $Q_k(Z)$ and $H_k$ are given by

$$\mathcal{E}(Z) = \frac{1}{2}\|\mathcal{R}_\Omega(AZB^\top - M)\|_F^2, \tag{16}$$

$$Q_k(Z) = \frac{L}{2}\left\|Z - \left(Y_k - \frac{1}{L}A^\top\mathcal{R}_\Omega(AY_kB - M)B\right)\right\|_F^2, \tag{17}$$

$$H_k(L) = \text{Tr}((Z_{k+1} - Y_k)^\top A^\top\mathcal{R}_\Omega(AY_kB^\top - M)) + \frac{L}{2}\|Z_{k+1} - Y_k\|_F^2. \tag{18}$$

It is based on the accelerated gradient descent method [5] that achieved a convergence of $O(1/T^2)$, where $T$ is the number of iterations, by explicitly exploiting the smoothness of the objective function. The stopping criteria $\epsilon$ is set to be a small constant. Besides the variable $Z$, Algorithm 1 introduces an auxiliary variable $Y$, which is updated based on a linear combination of $Z_k$ and $Z_{k-1}$ (Line 4). The singular value thresholding method [1] is used to solve $Z_{k+1} = \arg\min_Z \lambda\|Z\|_{tr} + Q_k(Z)$ (Line 5 and Line 8). Finally, instead of using the estimated upper bound for the smoothness of the objective function, which tends to be loss in practice, Algorithm 1 finds the best smoothness constant $L$ for the objective function by performing a line search (line 6-9) that terminates till the condition $\|\mathcal{R}_\Omega(AZ_{k+1}B^\top - M)\|_F^2 - \|\mathcal{R}_\Omega(AY_kB^\top - M)\|_F^2 \leq H_k(L)$ is satisfied. This idea was originally proposed in [3] and was adopted in [2, 5] to speed up the convergence of matrix completion.

## 3 Additional Experiments

Here the settings are the same as Section 4.2, but we provide the Average Precision measured on the whole data, instead of only on test instances. The results are provided in Table 3. We can see that our proposal either gets the best result, or it is comparable with the best result. Our proposal again achieves the best result on two big data sets, saying, NUS-WIDE and Flickr.