[Reviews · NeurIPS 2013]

Submitted by Assigned_Reviewer_3

The paper describes a method of completing an n-by-m matrix M with rank r smaller than min(n,m). Instead of completing M directly, the authors assume access to an n-by-r1 matrix A and an m-by-r2 matrix B such that one can write M=AZB'. The A, B matrices are considered as side information given beforehand and the proposed method completes M by estimating Z.

The authors prove that, when r1 is much smaller than n and r2 is much smaller than m, one needs far fewer entries to complete M by using the side information matrices. Moreover, in this case, Z is of much smaller dimensions than M, and thus the estimation is very efficient.

Matrix completion with side information has been considered previously in (Goldberg et. al., 2010; Adams, Dahl, andMurray, 2010; Porteous, Asuncion, and Welling, 2010). The method in this paper appears different from the previous ones, and the advantage over (Goldberg et. al., 2010) is shown in the paper.

The sample complexity analysis is nice and the experimental results are promising.

A negative feeling about the paper is, the side information assumption is a bit strong, and the reduced sample complexity and performance gain is therefore not so surprising. Nevertheless, the paper showed an example, where such an assumption is appropriate.



Summary: The paper presents a theoretical result showing that using side information matrices can significantly reduce sample complexity in matrix completion. The use of side information leads to an efficient matrix-completion method, which achieves speed-ups by estimating a low-dimensional matrix.

Submitted by Assigned_Reviewer_4

The paper studies the matrix completion problem in the context of collaborative filtering or multi-label learning. The contribution over existing work is that the authors account for side information on the rows and columns of the matrix in their model and show how exploiting side information can bring about improvements in sample complexity over the standard matrix completion formulation. In particular the authors study a model for matrix completion where the matrix of interest M can be decomposed as M = A Z_0 B^T with A and B is the known side information . When A and B are low dimensional this formulation has fewer degrees of freedom since the unknown matrix Z_0 is much smaller than M itself which intuitively explains why one should expect better sample complexity guarantees.

The authors analyze a very natural extension of the usual nuclear norm minimization program. They show that the sample complexity required to guarantee unique minimizer is O(\mu^2 r (r_a + r_b) \log n) for a n \times n matrix of rank r with Z_0 of dimension r_a \times r_b and where \mu is related to the usual incoherence parameter used in the literature. This implies that if the side information is low dimensional (r_a, r_b small) the sample complexity is sublinear in the matrix dimensions, which is an improvement over standard results.

The authors complement the theoretical results with simulations and experiments in the context of multi-label learning. The simulations show that their algorithm is both computationally and statistically more efficient that singular value thresholding, one of the standard matrix completion algorithms. Their algorithm also outperforms a number of baselines on several data sets in the multi-label learning application.

The paper studies an interesting problem and contributes to the body of work on matrix completion by capturing side information. The results are interesting and the paper is well written. The proof techniques are novel but not particularly innovative given the work on matrix completion. In light of this, I think the contribution is somewhat small. In particular, it would be nice to know the necessary conditions in the noiseless case and the guarantees for the noisy version of the matrix completion problem. The experiments are fairly convincing but could be presented better. Graphs in lieu of tables would lead to improved readability while also saving space.

Questions:
1. In the main paper there is no discussion about the role of \Omega_0 and \Omega_1 or any discussion about the condition \Omega_1 \ge q \Omega_0. It would be nice to provide some intuition as to why such a condition necessary and regimes under which it is satisfied.
2. Lemma 4 in the appendix requires that \Omega_0 \le |\Omega| \le \Omega_1, but in the main theorem there is no upper bound on the sample complexity. This seems like an error, but I'm more concerned with why there is an upper bound on |\Omega| in the first place. Is this condition necessary? What is the intuition for it?
3. Why are there test instances held out of the experiments? Why not just randomly sample the label matrix and test on the unobserved entries of that matrix?
4. The paper title is misleading -- Speedup implies computationally, which was not the focus of the paper. I read the paper thinking it would be much more of an algorithmic paper than it was.
Summary: The paper makes an interesting if small contribution to the matrix completion literature by studying one way to incorporate side information into the model and showing how this can lead to improved sample complexity guarantees. The proofs use what are now standard tools in convex analysis so while the formulation is novel, the analysis does not offer much technical innovation.

Submitted by Assigned_Reviewer_5

The paper develops new theories that use side information to reduce sample complexity, and demonstrates the approach's effectiveness in transductive multi-label learning.

For the proof, it is assumed that both A and B are orthonormal matrices (top paragraph in page 4). It would be better to clarify how the results will change when A and B are not orthogonal. Also since U and V are linear combinations of column vectors in A and B, the proof may be simplified.

The unconstrained optimization problem (3) is very similar to the problem solved in [1]. Actually, if we index the set of observation \Omega from 1 to k such that $\Omega_k = (i_k, j_k)$, and let $C_k = A_{i_k, .} B_{j_k, .}^T$. The problem (3) becomes $min_Z \sum_k \|M_k - tr( Z C_k)\|^2+ \lambda |Z|_{tr}$ which is exactly the problem solved in [1]. Similarly, [1] also provides consistency results and an efficient algorithm. It would be useful to compare to the algorithm for multi-label learning experiment.

In the end of second paragraph, it is said that recent efforts try to address the issue but at a price of losing performance guarantee. It's not clear as paper [5] cited in the original paper gives convergence guarantee. Actually there are several papers that provide efficient algorithms that don't require updating full matrix and also have convergence guarantees such as paper [2] below.

In proof of lemma 1 in supplementary material, there is a typo in line 063 that it should be $(1 - ||P_{T^{\perp}||)$ where the $\perp$ is missed.

[1] Francis R. Bach, Consistency of Trace Norm Minimization, JMLR 2008.
[2] M. Jaggi, M. Sulovsky. A Simple Algorithm for Nuclear Norm Regularized Problems, ICML 2010
Summary: The authors provide new theoretical and empirical results of sample complexity for matrix completion. The results should be compared to Bach(2008) that analyzed a very similar problem.
Author Feedback

Author rebuttal: 